# Immune dynamics in SARS-CoV-2 experienced immunosuppressed rheumatoid arthritis or multiple sclerosis patients vaccinated with mRNA-1273

Niels JM Verstegen[1], Ruth R Hagen[2,3†], Jet van den Dijssel[2,3†], Lisan H Kuijper[1†], Christine Kreher[1†], Thomas Ashhurst[4,5†], Laura YL Kummer[6], Maurice Steenhuis[1], Mariel Duurland[1], Rivka de Jongh[1], Nina de Jong[1], C Ellen van der Schoot[3], Amélie V Bos[1], Erik Mul[7], Katherine Kedzierska[8,9], Koos PJ van Dam[6], Eileen W Stalman[6], Laura Boekel[10], Gertjan Wolbink[1,10], Sander W Tas[11], Joep Killestein[12], Zoé LE van Kempen[12], Luuk Wieske[6,13], Taco W Kuijpers[14], Filip Eftimov[6], Theo Rispens[1], S Marieke van Ham[1,15‡], Anja ten Brinke[1*‡], Carolien E van de Sandt[2,8*‡], On behalf of the T2B! immunity against SARS-CoV-2 study group

[1]Department of Immunopathology, Sanquin Research and Landsteiner Laboratory, University of Amsterdam, Amsterdam, Netherlands; [2]Department of Hematopoiesis, Sanquin Research and Landsteiner Laboratory, University of Amsterdam, Amsterdam, Netherlands; [3]Department of Experimental Immunohematology, Sanquin Research and Landsteiner Laboratory, Amsterdam, Netherlands; [4]Sydney Cytometry Core Research Facility, Charles Perkins Centre, Centenary Institute, and The University of Sydney, Sydney, Australia; [5]School of Medical Sciences, Faculty of Medicine and Health, The University of Sydney, Sydney, Australia; [6]Department of Neurology and Neurophysiology, Amsterdam Neuroscience, University of Amsterdam, Amsterdam, Netherlands; [7]Department of Research Facilities, Sanquin Research, Amsterdam, Netherlands; [8]Department of Microbiology and Immunology, University of Melbourne at the Peter Doherty Institute for Infection and Immunity, Melbourne, Australia; [9]Global Station for Zoonosis Control, Global Institution for Collaborative Research and Education (GI-CoRE), Hokkaido University, Sapporo, Japan; [10]Department of Rheumatology, Amsterdam Rheumatology and immunology Center, Amsterdam, Netherlands; [11]Amsterdam Rheumatology and immunology Center, Department of Rheumatology and Clinical Immunology, University of Amsterdam, Amsterdam, Netherlands; [12]Amsterdam UMC, Vrije Universiteit, Department of Neurology, Amsterdam, Netherlands; [13]Department of Clinical Neurophysiology, St Antonius Hospital, Nieuwegein, Netherlands; [14]Department of Pediatric Immunology, Rheumatology and Infectious Disease, University of Amsterdam, Amsterdam, Netherlands; [15]Swammerdam Institute for Life Sciences, University of Amsterdam, Amsterdam, Netherlands

*For correspondence:
a.tenbrinke@sanquin.nl (AB);
cvandesandt@unimelb.edu.au
(CEvdS)

†These authors contributed
equally to this work
‡These authors also contributed
equally to this work

Competing interest: See page
19

Reviewing Editor: Sarah Sasson,
The Kirby Institute UNSW,
Australia

## Abstract

**Background:** Patients affected by different types of autoimmune diseases, including common conditions such as multiple sclerosis (MS) and rheumatoid arthritis (RA), are often treated with immunosuppressants to suppress disease activity. It is not fully understood how the severe acute respiratory

syndrome coronavirus 2 (SARS-CoV-2)-specific humoral and cellular immunity induced by infection and/or upon vaccination is affected by immunosuppressants.

**Methods:** The dynamics of cellular immune reactivation upon vaccination of SARS-CoV-2 experienced MS patients treated with the humanized anti-CD20 monoclonal antibody ocrelizumab (OCR) and RA patients treated with methotrexate (MTX) monotherapy were analyzed at great depth via high-dimensional flow cytometry of whole blood samples upon vaccination with the SARS-CoV-2 mRNA-1273 (Moderna) vaccine. Longitudinal B and T cell immune responses were compared to SARS-CoV-2 experienced healthy controls (HCs) before and 7 days after the first and second vaccination.

**Results:** OCR-treated MS patients exhibit a preserved recall response of CD8$^+$ T central memory cells following first vaccination compared to HCs and a similar CD4$^+$ circulating T follicular helper 1 and T helper 1 dynamics, whereas humoral and B cell responses were strongly impaired resulting in absence of SARS-CoV-2-specific humoral immunity. MTX treatment significantly delayed antibody levels and B reactivation following the first vaccination, including sustained inhibition of overall reactivation marker dynamics of the responding CD4$^+$ and CD8$^+$ T cells.

**Conclusions:** Together, these findings indicate that SARS-CoV-2 experienced MS-OCR patients may still benefit from vaccination by inducing a broad CD8$^+$ T cell response which has been associated with milder disease outcome. The delayed vaccine-induced IgG kinetics in RA-MTX patients indicate an increased risk after the first vaccination, which might require additional shielding or alternative strategies such as treatment interruptions in vulnerable patients.

**Funding:** This research project was supported by ZonMw (The Netherlands Organization for Health Research and Development, #10430072010007), the European Union's Horizon 2020 research and innovation program under the Marie Skłodowska-Curie grant agreement (#792532 and #860003), the European Commission (SUPPORT-E, #101015756) and by PPOC (#20_21 L2506), the NHMRC Leadership Investigator Grant (#1173871).

## Editor's evaluation

The article by Verstegen et al. examines humoral and cellular immune response in two subgroups of SARS-CoV-2-experienced immunosuppressed patients receiving two doses of mRNA-1273 vaccine. Further understanding the barriers to seroconversion to COVID-19 vaccination in immunosuppressed populations and how best to overcome these challenges are of great importance. The report is well written, logical, focused and thorough.

## Introduction

Severe acute respiratory syndrome coronavirus 2 (SARS-CoV-2), the virus that causes coronavirus disease 2019 (COVID-19), has infected millions of individuals, resulted in >4 million deaths, and greatly disrupted societies worldwide (*Dong et al., 2020*). Recovered individuals mostly exhibit robust humoral and cellular SARS-CoV-2 immunity (*Grifoni et al., 2020*; *Koutsakos et al., 2021*; *Mathew et al., 2020*; *Sekine et al., 2020*; *Thevarajan et al., 2020*) and generate immunological memory. In addition, multiple effective vaccines have been deployed to induce immune-mediated protection against SARS-CoV-2 leading to a substantial risk reduction of developing severe disease. Several studies in healthy individuals demonstrated that the mRNA vaccines, BNT162b2 (Pfizer-BioNTech) and mRNA-1273 (Moderna), successfully induce SARS-CoV-2-specific humoral (*Gaebler et al., 2021*; *Ma et al., 2020*) and cellular immunity (*Collier et al., 2021*; *Minervina et al., 2021*; *Oberhardt et al., 2021*; *Sahin et al., 2021*) and are highly efficacious (>94%) in reducing transmissibility and induction of serious disease and hospitalization of most variants (*Baden et al., 2021*; *Lyngse et al., 2021*; *Polack et al., 2020*; *Tenforde and Patel, 2021*). In addition to long-lived antibodies, the efficacy of these vaccines greatly relies on the induction of memory B and T cells.

However, about 4% of the world's population is affected by one of over 80 different types of autoimmune disease, including common conditions such as multiple sclerosis (MS) and rheumatoid arthritis (RA) (*National Stem Cell Foundation, 2021*). Autoimmune diseases are often treated with immunosuppressants which successfully suppress autoreactive immune responses (*Burmester and Pope, 2017*; *Cronstein and Aune, 2020*; *Edwards et al., 2004*; *Edwards and Cambridge, 2006*;

*Mulero et al., 2018*; *Wolinsky et al., 2020*), but simultaneously also affect the functionality of the adaptive immune system during infection or vaccination, depending on the immunosuppressant that is used. The severity of COVID-19 in individuals on immunosuppressants varies from mild to severe, depending on the type of immunosuppressants (*Gianfrancesco et al., 2020*; *Möhn et al., 2020*; *Sparks et al., 2021*; *Strangfeld et al., 2021*) and other underlying risk factors (*Baden et al., 2021*; *Chaudhry et al., 2021*; *Hughes et al., 2021*; *Möhn et al., 2020*; *Pablos et al., 2020*; *Sparks et al., 2021*; *Strangfeld et al., 2021*; *Williamson et al., 2020*; *Zabalza et al., 2021*). It is not well understood how immunosuppressants affect the formation of protective immunological B and T cell memory upon SARS-CoV-2 infection or vaccination, in part because these patient groups were excluded from many phase 3 clinical trials. Failure to induce effective immunological memory could leave these patients at risk for symptomatic re-infections. As SARS-CoV-2-specific immunity may not be very long-lived and/or can be breached by novel variants, like omicron that has mutated antibody recognition sites, it is important to evaluate if treatment with immunosuppressants allows efficient induction of SARS-CoV-2-specific B and broad-protective T cell memory upon SARS-CoV-2 exposure and subsequent reactivation of memory cells upon SARS-CoV-2 antigen recall (*Apostolidis et al., 2021*; *Felten et al., 2022*; *Goel et al., 2022*; *Haberman et al., 2022*; *Mahil et al., 2021*). To assess the latter, ex vivo evaluation of SARS-CoV-2 immune dynamics in patients with autoimmune diseases treated with immunosuppressants following recovery of SARS-CoV-2 infection and subsequent vaccination is required. Recent studies demonstrated that a single dose of mRNA vaccine in SARS-CoV-2 experienced healthy individuals resulted in stronger and broader immune responses than with vaccination alone, the so-called hybrid immunity (*Reynolds et al., 2021*; *Stamatatos et al., 2021*; *Tauzin et al., 2021*; *Wang et al., 2021a*). It remains to be established whether SARS-CoV-2 experienced individuals treated with immunosuppressants also induce a similar recall response upon vaccination. In this study, we aim to understand if and how immunosuppressive medication may interfere with the formation of protective immunological memory against SARS-CoV-2 to refine and/or optimize vaccine strategies to generate long-lasting, protective immunological memory in individuals using immunosuppressants.

B cell-depleting anti-CD20 monoclonal antibody (anti-CD20) therapies such as ocrelizumab (OCR) and rituximab (RTX) are successfully used to treat multiple diseases including MS (*Edwards et al., 2004*; *Edwards and Cambridge, 2006*; *Mulero et al., 2018*; *Wolinsky et al., 2020*). However, B cells play a critical role in the formation of protective humoral immunity during viral infections. With help from cognate CD4$^+$ T follicular helper cells (T$_{fh}$ cells), B cells differentiate into memory B cells and plasmablasts or long-lived plasma cells that secrete class-switched, high-affinity antibodies upon antigen stimulation through infection or vaccination (*Elsner and Shlomchik, 2020*; *Nutt et al., 2015*; *Verstegen et al., 2021*). SARS-CoV-2 infection in otherwise healthy individuals has demonstrated that both B and T$_{fh}$ cells are of great importance for the formation of a protective humoral immunity (*Koutsakos et al., 2021*; *Mathew et al., 2020*; *Ng et al., 2020*; *Thevarajan et al., 2020*; *Wang et al., 2021b*). However, the majority of patients on anti-CD20 therapies did not generate significant SARS-CoV-2-specific antibody titers following the natural infection (*Koutsakos et al., 2021*; *Simon et al., 2021*; *Zabalza et al., 2021*) or vaccination (*Apostolidis et al., 2021*; *Deepak et al., 2021*; *van Kempen et al., 2022*). A study in SARS-CoV-2 naïve anti-CD20-treated MS patients demonstrated that low vaccination-induced SARS-CoV-2 antibody titers correlate with reduced frequencies of circulating B cells (in line with anti-CD20-mediated B cell depletion) and low T$_{fh}$ cells (*Apostolidis et al., 2021*). Interestingly, SARS-CoV-2 naïve anti-CD20-treated MS patients which induced the lowest antibody titers displayed the highest induction of activated CD8$^+$ T cells after vaccination (*Apostolidis et al., 2021*; *Brill et al., 2021*; *Gadani et al., 2021*; *Madelon et al., 2021*). CD8$^+$ T cells form an important second line of protection against severe illness and mortality, as they are essential for the viral clearance (*Kundu et al., 2022*; *Tan et al., 2021*). To date, it remains to be evaluated if patients treated with anti-CD20 induce enough immunological B and T cell memory upon primary infection to be efficiently recalled upon re-exposure and/or with targeted vaccination.

Methotrexate (MTX) is another widely used immunosuppressant, which is one of the most common and effective medications to treat RA (*Burmester and Pope, 2017*; *Cronstein and Aune, 2020*). MTX has broad immunomodulating functions affecting multiple arms of the immune system and has been shown to reduce circulating leukocytes numbers (neutrophils and lymphocytes), their proliferation capacity, T cell receptor (TCR) activation, T cell lytic capacity, inhibiting pro-inflammatory pathways, neutrophil recruitment, extracellular trap formation and cytokine expression by macrophages and

increasing the number of regulatory T cells (T$_{regs}$) (*Cronstein and Aune, 2020*). Although the immune-modulatory function of MTX effectively mitigates effects of autoimmune inflammatory reactions and potentially contributes to reduced severity of COVID-19 (*Sparks et al., 2021*; *Strangfeld et al., 2021*), it has also been reported to reduce antibody titers after SARS-CoV-2 vaccination in SARS-CoV-2 naïve autoimmune patients (*Deepak et al., 2021*; *Furer et al., 2021*; *Haberman et al., 2020*; *Spiera et al., 2021*). In contrast to anti-CD20-treated MS patients, mRNA vaccination of SARS-CoV-2 naïve MTX-treated patients with immune-mediated inflammatory disease (IMID) did not induce activated CD8$^+$ T cells following mRNA vaccination (*Haberman et al., 2020*). Proportions of spike-specific B cells, T$_{fh}$, and activated CD4$^+$ T cells were reported to be induced to similar levels as observed in healthy individuals and IMID patients without MTX treatment although induction of antibody titers was less (*Haberman et al., 2020*). However, it is currently unknown whether MTX affects immunological memory formation after natural SARS-CoV-2 infection and/or B and T cell reactivation upon re-infection and/or vaccination.

To understand how immunosuppressive medications like OCR and MTX affect the induction of immunological memory and subsequent recall response, the humoral and cellular immune responses in SARS-CoV-2 experienced RA-MTX and MS-OCR patients were established and compared to SARS-CoV-2 experienced healthy individuals following the first and second dose of the Moderna mRNA vaccine. The immune dynamics of the recall response were elucidated by measuring SARS-CoV-2-specific antibody responses and deep-immune profiling of B and T cell responses in fresh peripheral blood. This study shows that OCR-treated MS patients exhibit a preserved recall response of CD8$^+$ T cells following first vaccination compared to healthy controls (HCs) and a normal CD4$^+$ T$_{fh1}$ and T$_{h1}$ dynamics, whereas humoral and B cell responses were strongly impaired. In contrast, RA patients treated with MTX displayed a delayed induction of the humoral recall responses after the first vaccination, but antibody levels were comparable to HCs following a second vaccination. However, MTX treatment delayed and/or hampered CD4$^+$ and CD8$^+$ T cell reactivation as demonstrated by the absence of co-expression of multiple dynamic markers.

# Materials and methods
## Study participants and design
Deep cellular immunological analysis was performed after SARS-CoV-2 vaccination as part of a national prospective longitudinal multi-arm multicenter cohort study focusing on the humoral response after SARS-CoV-2 vaccination in patients with the autoimmune disease treated with specific immunosuppressive medications. The study design and methods have been previously described (*Wieske et al., 2022*). This study was approved by the medical ethical committee (NL74974.018.20 and EudraCT 2021-001102-30, local METC number: 2020_194) and registered at Dutch Trial Register (Trial ID NL8900). Written informed consent was obtained from all study participants when enrolled. Participants were recruited between April 16, 2021, and May 20, 2021, at the MS Center Amsterdam, Amsterdam UMC, and the Amsterdam READE Rheumatology and Immunology Center and vaccinated between April 19, 2021, and July 1, 2021, with the mRNA-1273 (Moderna) vaccine at an interval of six weeks, according to the Dutch national vaccination guidelines. Peripheral blood was collected by venipuncture directly before the first vaccination (T0), 7–10 days after the first vaccination (T1), and 7–10 days after the second vaccination (T3). Antibody responses were measured directly before the first vaccination (T0), 7–10 (T1), and 42 days (T2) post first vaccination, and 7–10 (T3) and 28–56 days (T4) post second vaccination.

Included participants, aged ≥18 years, were eligible for vaccination conform to the Dutch national vaccination campaign. We included participants diagnosed with relapsing-remitting multiple sclerosis using OCR (MS-OCR) and participants with rheumatoid arthritis using methotrexate (RA-MTX). Participants had been diagnosed with relapsing-remitting MS and RA by a neurologist or rheumatologist, respectively. In addition, we recruited control participants (HC) who had no history of an immune-mediated disorder and did not use any form of systemic immunosuppressive therapy. All study participants had a prior SARS-CoV-2 infection proven by RT-PCR or antibody test. Exclusion criteria were immunosuppressive co-medication, incorrect diagnosis, withdrawal from informed consent, and a positive SARS-CoV-2 test less than 8 weeks before vaccination (*Figure 1a*). Baseline characteristics including comorbidities were collected from all study participants (*Supplementary file 1*). The

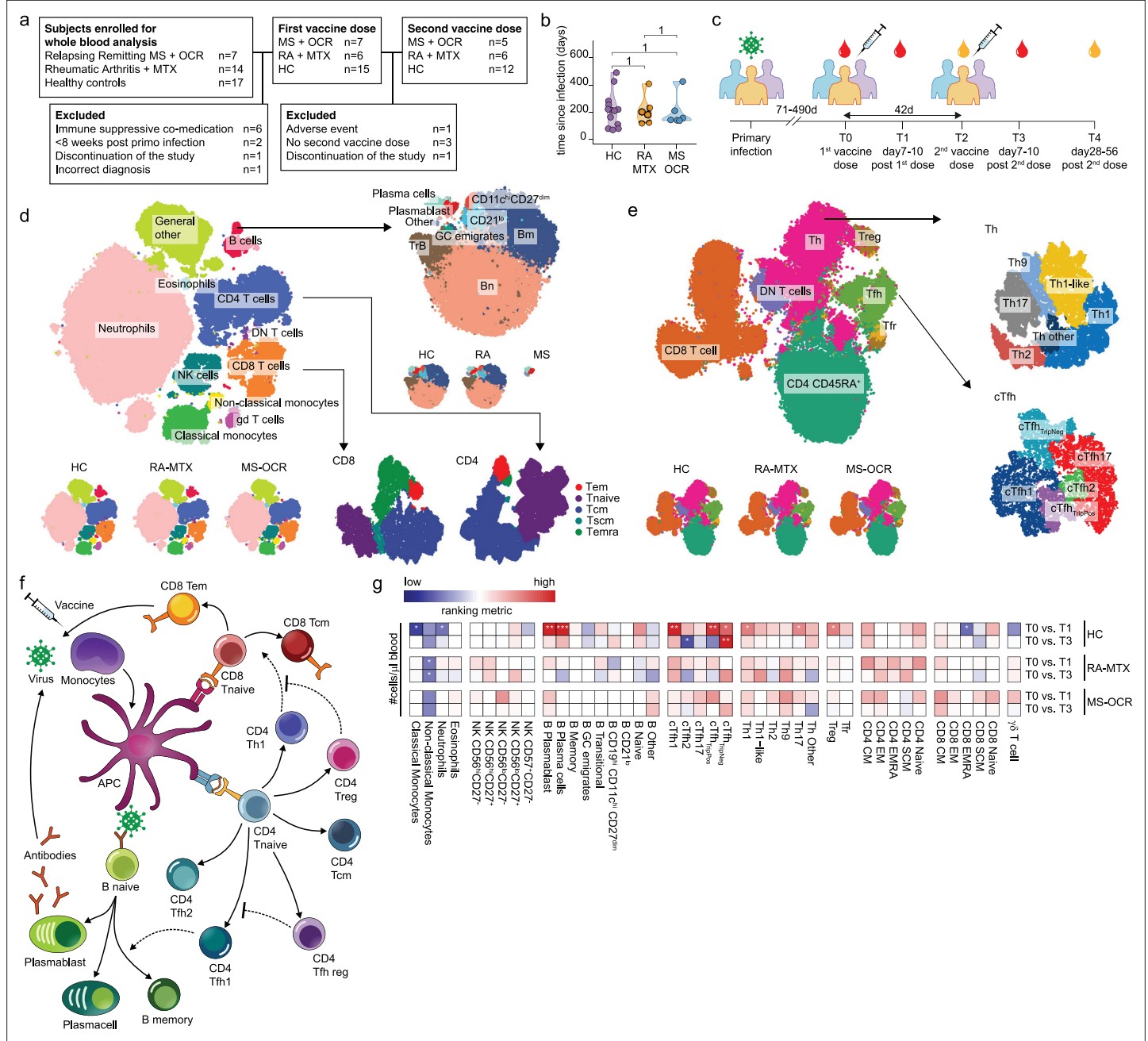

**Figure 1.** Dynamics of circulating immune populations after severe acute respiratory syndrome coronavirus 2 (SARS-CoV-2) mRNA vaccination.
(**a**) Overview of cohort (**b**) distribution of time since primary infection within the different groups and (**c**) longitudinal study design. SARS-CoV-2 mRNA vaccination was administrated in two doses to SARS-CoV-2 experienced RA patients on methotrexate (RA-MTX) treatment, MS patients on ocrelizumab (MS-OCR) treatment, and healthy controls (HCs). Whole blood (red) or serum (yellow) was collected at indicated time points. (**d–e**) UMAP and cluster identification from FlowSOM analysis of two high-dimensional flow cytometry panels. The UMAP is separated across groups and major adaptive immune populations were subclustered in individual projections. (**f**) Schematic overview of circulating immune populations and their interactions. (**g**) Representation of cell count of immune populations that are affected by first (**T1**) and second (**T3**) vaccination as compared to baseline (**T0**) in the different groups. Each immune population is represented by a single column of colored squares, and each time point-disease group combination is represented by a single row. Color squares represent the populations that are reduced (blue), increased (red), or not affected (white). Statistical significance was determined using a Wilcoxon signed-rank test with Bonferroni-Holm multiple comparison correction.

The online version of this article includes the following figure supplement(s) for figure 1:

**Figure supplement 1.** Spectre analysis of circulating immune populations after severe acute respiratory syndrome coronavirus 2 (SARS-CoV-2) mRNA vaccination.

*Figure 1 continued on next page*

*Figure 1 continued*

**Figure supplement 2.** Circulating immune populations after severe acute respiratory syndrome coronavirus 2 (SARS-CoV-2) mRNA vaccination of SARS-CoV-2 experienced rheumatoid arthritis using methotrexate (RA-MTX), multiple sclerosis using ocrelizumab (MS-OCR), and healthy control (HC).

**Figure supplement 3.** Innate immune cells in the blood of severe acute respiratory syndrome coronavirus 2 (SARS-CoV-2) mRNA vaccinated SARS-CoV-2 experienced rheumatoid arthritis using methotrexate (RA-MTX) and multiple sclerosis using ocrelizumab (MS-OCR) patients.

severity of SARS-CoV-2 primary infection was defined as asymptomatic (no clinical symptoms), mild (no hospitalization), moderated (hospitalization), and severe (ICU admission). OCR therapy was scored as the time between last infusion and primary infection/first vaccination, and the number of past infusions. All RA participants used MTX at a stable dose for at least 6 months before SARS-CoV-2 primary infection.

## Whole blood flow cytometry

Fresh whole blood was used to measure cellular immune populations, essentially as described in *Koutsakos et al., 2021*; *Thevarajan et al., 2020*, using two human antibody panels (*Supplementary file 2*). Fresh whole blood (200 µl) was stained with the respective panel for 30 min at room temperature (RT) in the dark. Next, samples were lysed with BD FACS Lysing solution (BD Biosciences) for 10 min at RT in the dark, washed and subsequently fixed with 1% PFA for 20 min at 4°C in the dark, washed, and resuspended in PBS supplemented with 0.5% bovine serum albumin and 2 mM ethylenediaminetetraacetic acid. Count bright Plus Absolute Counting Beads (ThermoFisher) were added for calculating absolute cell numbers just before acquisition. Samples were acquired on a BD FACSymphony (BD).

## RBD protein ELISAs

Detection of IgG, IgM, and IgA directed to RBD were measured as described previously (*Steenhuis et al., 2021*; *Vogelzang et al., 2020*). RBD proteins were produced as described previously (*Vogelzang et al., 2020*). In short, MaxiSorp microtiter plates (Thermo Fisher Scientific) were coated with 1.0 mg/ml RBD-ST or 4.0 mg/ml monoclonal mouse anti-human IgM (MH15; Sanquin) in PBS overnight at 4°C. Subsequently, plates were washed with PBS-T and plasma samples were incubated for 1 hr at RT. After washing, 0.5 mg/ml HRP-conjugated monoclonal mouse anti-human IgG (MH16; Sanquin) or anti-human IgA (MH14; Sanquin) was added, diluted in PTG, and incubated for 1 hr. The ELISAs were developed with 100 µg/ml tetramethylbenzidine in 0.11 mol/l sodium acetate (pH 5.5) containing 0.003% (v/v) hydrogen peroxide (Merck). The reaction was stopped with 2 M $H_2SO_4$. Absorption at 450 and 540 nm was measured with a Synergy 2 microplate reader (Biotek, Winooski, VT). For IgM, the assay was finished using 0.5 mg/ml biotinylated RBD-ST (EZ-Link Sulfo-NHS-LC-Biotin; Thermo Fisher Scientific) in PTG and incubated for 1 hr at RT, followed by incubation for 30 min with streptavidin–poly-HRP (Sanquin). The antibody titers of the time points T0, T1, and T3 were determined in plasma (P) and for time point T2 in serum (S).

## Computational flow cytometry analysis

Computational analysis of data was performed using the Spectre R package (*Ashhurst et al., 2021*). Initially, samples were loaded in FlowJo v10 software (FlowJo) and cells were gated on single cells. Anomalies were then detected and removed using the flowAI R package (*Monaco et al., 2016*). An arcsinh transformation was performed, and data below the limit of detection were compressed to reduce the contribution of noise to the clustering process. To mitigate the presence of batch effects, samples were integrated using reciprocal principal component analysis (rPCA) from the Seurat toolkit for the cellular genomics (*Hao et al., 2021*), as implemented in Spectre. rPCA projects the data from one batch into the PCA space of another, where cells are then paired across datasets using a mutual nearest neighbor approach (*Hao et al., 2021*), allowing for normalization of expression levels. Here, we chose a single batch as the 'reference' batch, and integrated each other batch with the reference batch, to decrease the total runtime. For subset discovery, high-dimensional FlowSOM data analysis and visualization of flow cytometry data were performed with all the non-dynamic surface molecules as input. Cluster identities were annotated manually by three individuals independently: monocytes (FSC$^{int}$CD14$^+$), neutrophils (FSC$^{int}$SSC$^{int}$CD16$^{hi}$CD10$^{hi}$), eosinophils (FSC$^{hi}$SSC$^{hi}$CD16$^-$), natural killer (NK) cells (CD56$^+$), B cells (CD19$^+$), CD4 T cells (CD3$^+$CD4$^+$), CD8 T cells (CD3$^+$CD8$^+$) and gamma-delta

T cells (TCRgd$^+$CD3$^+$), circulating T follicular helper (cT$_{fh}$) cells (CD3$^+$CD4$^+$CXCR5$^+$), memory T helper (T$_h$) cells (CD3$^+$CD4$^+$CD45RA$^-$CXCR5$^-$). The main circulating immune populations were then manually annotated in subclusters based on marker expression: monocytes (classical CD14$^+$CD16$^-$ and non-classical CD14$^-$CD16$^+$), NK cells (CD56$^{hi}$CD27$^-$, CD56$^{hi}$CD27$^+$, CD56$^{lo}$CD27$^-$, CD56$^{lo}$CD27$^+$, CD57$^+$CD27$^-$), B cells (plasmablast CD27$^+$CD38$^+$CD138$^-$, plasma cells CD27$^+$CD38$^+$CD138$^+$, memory CD27$^{dim}$CD38$^-$, GC emigrates CD27$^{dim}$CD10$^+$, transitional CD38$^+$CD10$^+$, CD19$^{hi}$CD11c$^{hi}$CD27$^{dim}$, CD21$^{lo}$, naïve CD27$^-$CD38$^-$), CD4$^+$ T cells (central memory [CM] CD45RA$^-$CD27$^+$, effector memory [EM] CD45RA$^-$CD27$^-$, effector memory CD45RA$^+$ [EMRA] CD45RA$^+$CD27$^-$, stem cell memory CD45RA$^+$CD27$^+$CD95$^+$, naïve CD45RA$^+$CD27$^+$CD95$^-$), CD8$^+$ T cells (central memory [CM] CD45RA$^-$CD27$^+$, effector memory [EM] CD45RA$^-$CD27$^-$, effector memory CD45RA$^+$ [EMRA] CD45RA$^+$CD27$^-$, stem cell memory CD45RA$^+$CD27$^+$CD95$^+$, naïve CD45RA$^+$CD27$^+$CD95$^-$). Lymphocytes from the T cell activation/ exhaustion panel were clustered and plotted by UMAP and CD4$^+$ T cell subsets were manually annotated based on marker expression: cT$_{fh}$ cells (cT$_{fh1}$ CXCR3$^+$CCR6$^-$CCR4$^-$, cT$_{fh2}$ CXCR3$^-$CCR6$^-$CCR4$^+$, cT$_{fh17}$ CXCR3$^-$CCR6$^+$CCR4$^+$, cT$_{fhTriplePos}$ CXCR3$^+$CCR6$^+$CCR4$^+$, cT$_{fhTripleNeg}$ CXCR3$^-$CCR6$^-$CCR4$^-$), memory T$_h$ cells (T$_{h1}$ CXCR3$^+$CCR6$^-$CCR4$^-$, T$_{h1}$-like CXCR3$^+$CCR6$^+$CCR4$^+$, T$_{h2}$ CXCR3$^-$CCR6$^-$CCR4$^+$, T$_{h9}$ CXCR3$^-$CCR6$^+$CCR4$^-$, T$_{h17}$ CXCR3$^-$CCR6$^+$CCR4$^+$), T$_{reg}$ CD127$^-$CD25$^+$CXCR5$^-$, T follicular regulatory (T$_{fr}$) cells CD127$^-$CD25$^+$CXCR5$^+$.

## Statistical analyses

Summary statistics, (connected) violin plots, stacked plots, volcano plots, and heatmaps were created in R. Statistical significance was assessed using Mann-Whitney or Wilcoxon signed-rank test using wilcox.test function in R and p-values were corrected for multiple comparison using the Bonferroni-Holm method (*Holm, 1979*). Adjusted p-values lower than 0.05 were considered statistically significant. The ranking metric used in the heatmaps is a score that combines fold-change and p-value and was calculated using -log10({adjusted p-value}) * sign({log2 fold-change}) (*Xiao et al., 2014*).

## Results

### SARS-CoV-2 experienced autoimmune cohort and study design

In the Target to B study (T2B!), 38 SARS-CoV-2 experienced individuals >18 years of age were recruited, of which 28 were selected, based on in- and exclusion criteria, to participate in a prospective cohort study to perform deep-immune profiling and evaluate dynamics of B and T cell responses in fresh peripheral blood between May 14, 2021, and July 9, 2021 (*Wieske et al., 2022*). The cohort consisted of 7 OCR-treated relapsing-remitting MS (MS-OCR) patients, 6 MTX-treated RA patients (RA-MTX), and 15 HCs (*Figure 1a* and *Supplementary file 1*). The median age, sex, and time since SARS-CoV-2 infection (*Figure 1b*) were comparable between all three groups, although the RA-MTX group had a slightly higher prevalence of comorbidities (*Supplementary file 1*). Most patients, except for one MS and one RA patient, did not require hospital admission during COVID-19 (*Supplementary file 1*). Patients received their first mRNA-1273 (Moderna) vaccination after full recovery from their SARS-CoV-2 infection, on average 210 (71–490) days post infection. One MS patient who experienced a severe adverse event after the first vaccination was excluded from the study (*Figure 1a*). The second vaccination was given ~42 days later, except for three HCs who, by recommendations of the Dutch national vaccination guidelines, did not wish to receive a second dose and one MS patient who discontinued the study. These participants were excluded from further analyses (*Figure 1a* and *Supplementary file 1*).

### The breadth of the studied immune response in RA-MTX and MS-OCR patients

The effect of OCR and MTX on the dynamic of immunological recall responses was elucidated by measuring SARS-CoV-2-specific antibody responses and deep-immune profiling of cellular immune responses via high-dimensional flow cytometry analysis on fresh whole blood samples at baseline (T0), post first- (T1) and second vaccination (T3) and for antibodies, additionally in serum samples pre (T2) and post (T4) second vaccination (*Figure 1c*). Antibody secreting B cell populations, especially plasmablast and plasma cells, can only be studied with high accuracy in whole blood because these cells are highly vulnerable and may not survive freeze-thaw procedures. In total 38 unique

flow cytometry markers (**Supplementary file 2**) were combined with a computational pipeline in the Spectre R package (**Ashhurst et al., 2021**), encompassing the rPCA (from Seurat) (**Hao et al., 2021**), FlowSOM (**Van Gassen et al., 2015**), and Flt-SNE (fast Fourier transform-accelerated interpolation-based t-stochastic neighborhood embedding) (**Linderman et al., 2019**) algorithms resulting in the identification of clusters representing major myeloid, innate, and lymphocyte lineages and their phenotype and activation status (**Figure 1d–e** and **Figure 1—figure supplement 1a, b**). In total, 42 different cell populations were identified, of which many function in complex interplay to combat viral infection (**Figure 1f–g**). A T cell activation panel was used to define clusters of memory CD4$^+$ T cell subsets, CD8$^+$ T cell phenotypes, and more in-depth analysis of dynamics markers previously associated with activated/responding T cells (**Ellebedy et al., 2016**; **Geers et al., 2021**; **Habel et al., 2020**; **Koutsakos et al., 2021**; **McElroy et al., 2015**; **Mudd et al., 2022**; **Nguyen et al., 2021**; **Oja et al., 2020**; **Rha et al., 2021**; **Thevarajan et al., 2020**), namely CD38, HLA-DR, PD-1, CTLA-4, TIGIT, TIM-3, CD40L ,and CD137 (**Figure 1e** and **Figure 1—figure supplement 1c-e**). The immunophenotype analysis demonstrated an absence of B cell populations which were largely depleted by OCR treatment in MS patients (**Figure 1d**). Interestingly, circulating plasmablast and plasma cells could still be observed in line with the absence of CD20 expression in these cell types (**Figure 1d**).

## Different immune dynamics between first and second vaccination and between patient groups

To assess recall immune profiles in samples from SARS-CoV-2 experienced MS-OCR, RA-MTX, and HC following first and second COVID-19 vaccination, cell numbers per μl blood, and percentages of the 42 immune populations identified by FlowSOM were compared between the groups. Significant changes were observed in multiple immune populations following the first and second vaccination (**Figure 1g** and **Figure 1—figure supplement 2a-c**). Superior immune reactivity was observed in HC showing significantly larger antibody-secreting cell populations upon first and not second vaccination as well as significant changes in cT$_{fh}$ and T$_h$ cells (**Figure 1g** and **Figure 1—figure supplement 2a-c**). Substantial changes in the size of various adaptive immune populations were observed in RA-MTX and MS-OCR (**Figure 1g**).

To establish whether vaccination affected innate immune responses 7–10 days after vaccination, we compared the dynamics of classical monocytes, non-classical monocytes, neutrophils, and NK subsets (**Figure 1g** and **Figure 1—figure supplement 3**). The number of non-classical monocytes per μl blood was significantly reduced in RA-MTX following the both first and second vaccination, whereas in HC and MS-OCR a substantial reduction was observed (**Figure 1g** and **Figure 1—figure supplement 3a-c**). In addition, the number of neutrophils and classical monocytes per μl blood were significantly reduced in HCs following the first vaccination (**Figure 1g** and **Figure 1—figure supplement 3a-c**). No profound changes were observed in the proportion of various subsets of NK cells in all groups following both vaccinations (**Figure 1—figure supplement 3d**). In contrast, an increased expression of the activation marker CD38 was observed on a broad range of NK subsets in all groups following both vaccinations as compared to the baseline sample (**Figure 1—figure supplement 3d**). However, we may have missed the peak of innate immune response, which is typically observed 1–2 days after vaccination.

In contrast to the limited innate immune responses following vaccination, more dynamic changes were observed in the adaptive immune profile of the three patient groups, which warrants further in-depth analysis.

## OCR and MTX treatment affect seroconversion and antibody recall dynamics

Convalescent humoral recall responses were analyzed by comparing RBD-specific IgG, IgM, and IgA antibodies between SARS-CoV-2 experienced HCs, RA-MTX, and MS-OCR patients before vaccination (T0) (**Figure 2a–c** and **Figure 2—figure supplement 1a, b**). Based on IgG titers, the majority of SARS-CoV-2 experienced RA-MTX (67%), MS-OCR patients (50%), and HC (58%) still had detectible antibody titers (antibody titers above the cutoff of 4 AU/ml levels; determined using 600 pre-COVID-19 outbreak samples as published before) (**Steenhuis et al., 2021**; **Vogelzang et al., 2020**; **Figure 2a** and **Figure 2—figure supplement 1a, b**). This indicates that autoimmune patients treated with MTX and OCR are capable of mounting SARS-CoV-2-specific humoral immunity following natural

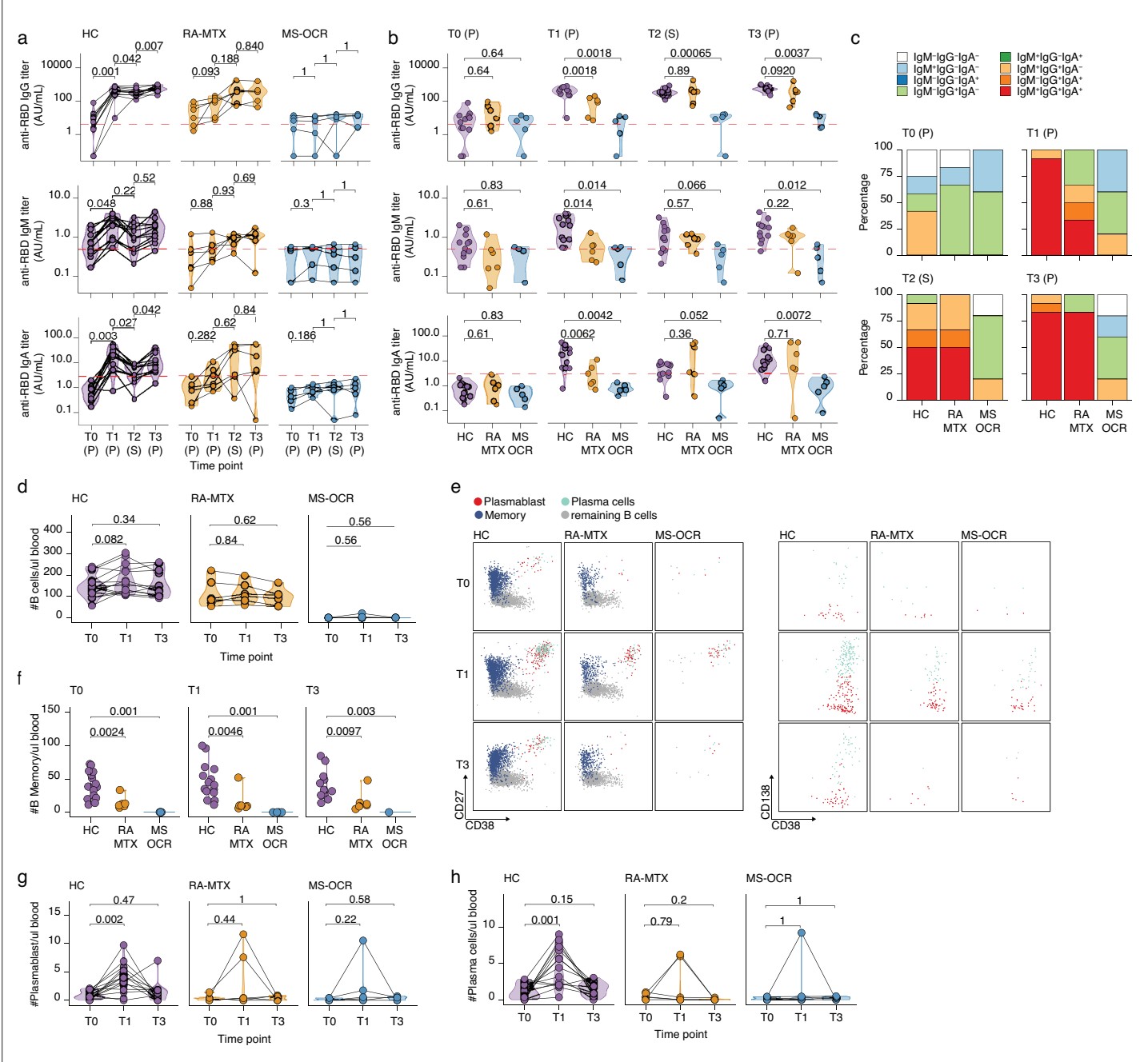

**Figure 2.** Antibody and circulating B cells responses of severe acute respiratory syndrome coronavirus 2 (SARS-CoV-2) mRNA vaccinated SARS-CoV-2 experienced rheumatoid arthritis using methotrexate (RA-MTX) and multiple sclerosis using ocrelizumab (MS-OCR) patients. (**a–c**) Results of ELISAs measuring antibody reactivity to RBD before vaccination (**T0**), 7 days after the first vaccination (**T1**), before second vaccination (**T2**), and 7 days after the second vaccination (**T3**) in SARS-CoV-2 experienced RA-MTX and MMS-OCR and healthy controls (HCs). The antibody titers of the time points T0, T1, and T3 were determined in plasma (**P**) and for time point T2 in serum (**S**). Anti-RBD IgG (top), anti-RBD IgM (middle), and anti-RBD IgA (bottom) levels are plotted longitudinally (**a**) or separated by groups across time points (**b**). (**c**) Percentage of participants seropositive for anti-RBD of the IgG, IgA, and/ or IgM isotypes. (**d**) Count per µl of blood of total B cells. (**e**) Representative flow cytometry plots for the quantification of circulating B cell populations. Colors represent the populations that were identified by unbiased analysis. (**f–h**) Count per µl of blood of memory B cells (**f**), plasmablast (**g**), and plasma cells (**h**) before (**T0**) and 7 days after first (**T1**) and second vaccination (**T3**). Statistical significance was determined using a Wilcoxon signed-rank test (a, d, g, and h) or a Mann-Whitney test (**b and f**) with Bonferroni-Holm multiple comparison correction.

The online version of this article includes the following figure supplement(s) for figure 2:

**Figure supplement 1.** Antibody and circulating B cells responses in severe acute respiratory syndrome coronavirus 2 (SARS-CoV-2) mRNA vaccinated SARS-CoV-2 experienced rheumatoid arthritis using methotrexate (RA-MTX) and multiple sclerosis using ocrelizumab (MS-OCR) patients.

SARS-CoV-2 infection. Furthermore, antibody responses can be maintained for prolonged periods comparable to healthy individuals in most MTX and OCR-treated patients, albeit at slightly lower levels in MS-OCR patients (*Figure 2b*). Seroconversion and isotype switching was especially surprising for the MS-OCR group considering their treatment with B cell-depleting medication before their PCR-proven SARS-CoV-2 infection, as has been shown previously (*Apostolidis et al., 2021*; *Moser et al., 2022*; *van Kempen et al., 2022*).

Vaccine-induced humoral recall responses were observed in SARS-CoV-2 experienced RA-MTX patients and HCs as 100% of the individuals seroconverted for IgG following the first vaccination (T1) and was maintained for the duration of the study ( Figure 4-figure supplement 1 b). However, the MTX treatment of RA patients had a major effect on the dynamics of the humoral immune response. Although RA-MTX patients displayed a slight increase in RBD IgG levels at day 7 post first vaccination (T1) (*Figure 2a*), they were significantly lower as compared to HC (*Figure 2b*). While by day 42 post first vaccination (T2), RA-MTX IgG levels had significantly increased to levels comparable to those observed in vaccinated HCs and a similar trend was observed for IgA and IgM (*Figure 2a–b*). Second vaccination of SARS-CoV-2 experienced RA-MTX patients did not result in a further increase of their antibody levels (T3) (*Figure 2a*) and levels remained comparable to vaccinated SARS-CoV-2 experienced HCs till at least 28–56 days post second vaccination (T4) (*Figure 2b* and *Figure 2—figure supplement 1b*). Together, this suggests a delayed SARS-CoV-2-specific antibody recall response in RA-MTX patients. No changes in antibody titers following both vaccinations were observed in SARS-CoV-2 experienced MS-OCR patients (*Figure 2a* and *Figure 2—figure supplement 1b*) and antibody levels for all isotypes remained significantly lower compared to vaccinated SARS-CoV-2 experienced HCs (*Figure 2b*). This indicates an impaired ability of SARS-CoV-2 experienced MS-OCR patients to recall the humoral immune response. In addition to variations in seroconversion and antibody levels in the MS-OCR group, a different isotype profile was observed compared to HCs (*Figure 2c*). Most notably, the combination of all three isotypes specific for RBD was not detected in any of the vaccinated SARS-CoV-2 experienced MS-OCR patients neither before nor after both vaccinations (*Figure 2c*). Although all three isotypes were detected in RA-MTX donors, the proportion was smaller as compared to HCs at T1 (HC 92%, RA-MTX 33.3%) and comparable pre- (T2; HC 50%, RA-MTX 50%) and post-second vaccination (T3; HC 83% RA-MTX 83%) (*Figure 2c*). Together, these data indicate that treatment with immunosuppressants MTX and OCR treatment resulted in different dynamics of the SARS-CoV-2-specific humoral recall immune response following vaccination of SARS-CoV-2 experienced RA and MS patients.

## Reduced B cell recall dynamics following vaccination in SARS-CoV-2 experienced RA-MTX and MS-OCR patients

Next, the dynamics of immune populations that underpin the induction of humoral immunity, namely B cells, were defined (*Figure 2d–h* and *Figure 2—figure supplement 1c, d*). While HC showed an expected increase in the number and proportion of total B cells at 7 days post first vaccination (T1), this was not observed in RA-MTX patients and B cells were almost absent in MS-OCR patients, in line with the B cell-depleting function of OCR (*Figure 2d*). Additional differences were observed when specialized subpopulations were analyzed (*Figure 2e–h* and *Figure 2—figure supplement 1c* , d), including a significantly lower number of memory B cells in SARS-CoV-2 experienced MS-OCR and RA-MTX patients compared to HCs at all three time points (T0, T1, and T3) (*Figure 2f* and *Figure 2—figure supplement 1c, d*). Interestingly, although two RA-MTX patients and one MS-OCR patient displayed an increase in plasmablast and plasma cell numbers and proportions (*Figure 2g–h* and *Figure 2—figure supplement 1d*), no increase in plasmablasts and plasma cells were observed in the majority of the SARS-CoV-2 experienced RA-MTX and MS-OCR patients upon vaccination (*Figure 2g–h*). A significantly higher number and proportion of plasmablast and plasma cells were observed following the first but not second vaccination of SARS-CoV-2 experienced HCs (*Figure 2g–h* and *Figure 2—figure supplement 1d*), consistent with the relatively small rise in antibody levels following the second vaccination (*Figure 2a*). Others have also reported a larger antibody-secreting cell (plasmablast and plasma cells) population following first vaccination in SARS-CoV-2 experienced versus naïve healthy individuals, with a reversed antibody-secreting cell response following second vaccination (*Samanovic et al., 2021*). Since maximum antibody levels in HCs were reached at 7–10 days after first vaccination and RA-MTX patients were capable of reaching similar levels but only 42 days after first vaccination

(*Figure 2a–b*), it seems that MTX treatment has mainly affected, potentially delayed, the dynamics of the B cell recall response following vaccination.

## RA-MTX and MS-OCR patients display a CD4$^+$ T$_{fh}$ cell recall response of different quality

T$_{fh}$ cells play a critical role in helping B cell activation and differentiation, antibody production, class switching, and somatic hypermutations which further strengthen overall antibody responses upon infection and vaccination (*Figure 1e*; *Koutsakos et al., 2019*). A small but significant increase in total CD4$^+$ circulating T$_{fh}$ cells per µl blood in HCs was observed following the first vaccination (*Figure 3a*). Through deep-immune profiling five CD4$^+$ cT$_{fh}$ subsets were identified (*Figure 1d*, *Figure 3b*). While the number and proportion of most CD4$^+$ cT$_{fh}$ subsets were remarkably similar between HC and both patients groups, a significantly smaller T$_{fh1}$ cell population was observed in RA-MTX and MS-OCR patients as compared to HC before vaccination (*Figure 3—figure supplement 1a, b*). This same population has been reported to correlate with more efficient B cells and humoral immunity after vaccination (*Koutsakos et al., 2019*). Interestingly, a significant increase in the number of CD4$^+$ cT$_{fh1}$, cT$_{fhTripPos}$, and cT$_{fhTripNeg}$ cells was determined upon first vaccination in HC, whereas this population remained unchanged in the patient groups (*Figure 3c*).

Next, the functional activation/exhaustion profile of the CD4$^+$ cT$_{fh}$ cells upon vaccination was assessed. Expression of CD38, HLA-DR, PD-1, CTLA-4, TIGIT, TIM-3, CD40L, and CD137 (*Figure 1—figure supplement 1b*) on CD4$^+$ cT$_{fh}$ subsets was used to verify activation upon vaccination. Only the proportion of CD38, HLA-DR, PD-1, CTLA-4, or TIGIT expressing cT$_{fh}$ cells significantly increased for one or several subsets (*Figure 3d*). Only limited overlap of these expression profiles was observed between SARS-CoV-2 experienced RA-MTX, MS-OCR patients and HCs following first and second vaccination (*Figure 3d*) indicating that both MTX and OCR affect the specific response profile of CD4$^+$ cT$_{fh}$ subsets. Analysis of the proportion of responding cells (cells that upregulated one or more dynamic markers) demonstrated that the CD4$^+$ cT$_{fh1}$, cT$_{fh17}$, and cT$_{fhTripPos}$ subsets are the most responsive after the first vaccination (*Figure 3e* and *Figure 3—figure supplement 1c*). The combined expression of these dynamic markers could be indicative of the quality of the response. A slightly higher combined expression of three dynamic markers on CD4$^+$ cT$_{fh1}$ cells following first vaccination was observed in MS-OCR patients compared to HCs, whereas the combined expression of multiple dynamic markers in RA-MTX patients was slightly lower compared to HCs (*Figure 3f* and *Figure 3—figure supplement 1d*). Overall, these data indicate that immunosuppressants MTX and OCR dampen a rise in total CD4$^+$ cT$_{fh}$ cell numbers per µl blood following the first vaccination, however, they hardly affected the responding CD4$^+$ cT$_{fh1}$ cell populations, although the response in MS-OCR patients was of slightly higher quality compared to those observed in HC and RA-MTX patients.

## Higher quality CD4$^+$ T$_{h1}$ recall response in MS-OCR patients

Next, the immune dynamics of the recall response of various T$_h$ cell effector memory subsets were investigated (*Figure 4a* and *Figure 4—figure supplement 1*). The number and proportion of T$_h$ was not different between all three groups before and after both vaccinations (*Figure 4a*). T$_{h1}$ and T$_{h1-like}$ subsets are major players during viral infections as they are known to promote CD8$^+$ T cell responses and are essential for the induction of memory CD8$^+$ T cells (*Deliyannis et al., 2002*; *Riberdy et al., 2000*; *Zhu and Paul, 2009*; *Figure 4b*). Although the size of the T$_{h1}$ population was comparable between HC and the patient groups, vaccination did promote the expansion of the T$_{h1}$ cell population in HC only, which resulted in significantly more T$_{h1}$ cells in HC after two vaccinations as compared to the patient groups (*Figure 4c–d* and *Figure 4—figure supplement 1a, b*). Furthermore, vaccination did not affect the size of the T$_{h1-like}$ population in the individual groups, however, after the first vaccination the size T$_{h1-like}$ cell population was significantly higher in HC where it only tended to be higher before vaccination (*Figure 4e–f* and *Figure 4—figure supplement 1a, b*).

Next, the response profile of CD4$^+$ T$_h$ subsets was analyzed using the same eight dynamic markers as described for cT$_{fh}$ cells as a surrogate for antigen-induced recall (*Figure 1—figure supplement 1b*). Although all eight dynamic markers were significantly acquired by all of the CD4$^+$ T$_h$ subsets following the first and/or second vaccination, activation of the T$_{h1}$ subset was most superior (*Figure 4g* and *Figure 4—figure supplement 1c*). A profound increase in responding CD4$^+$ T$_{h1}$ cells was observed in all SARS-CoV-2 experienced groups after the first vaccination, which was significant for the HC

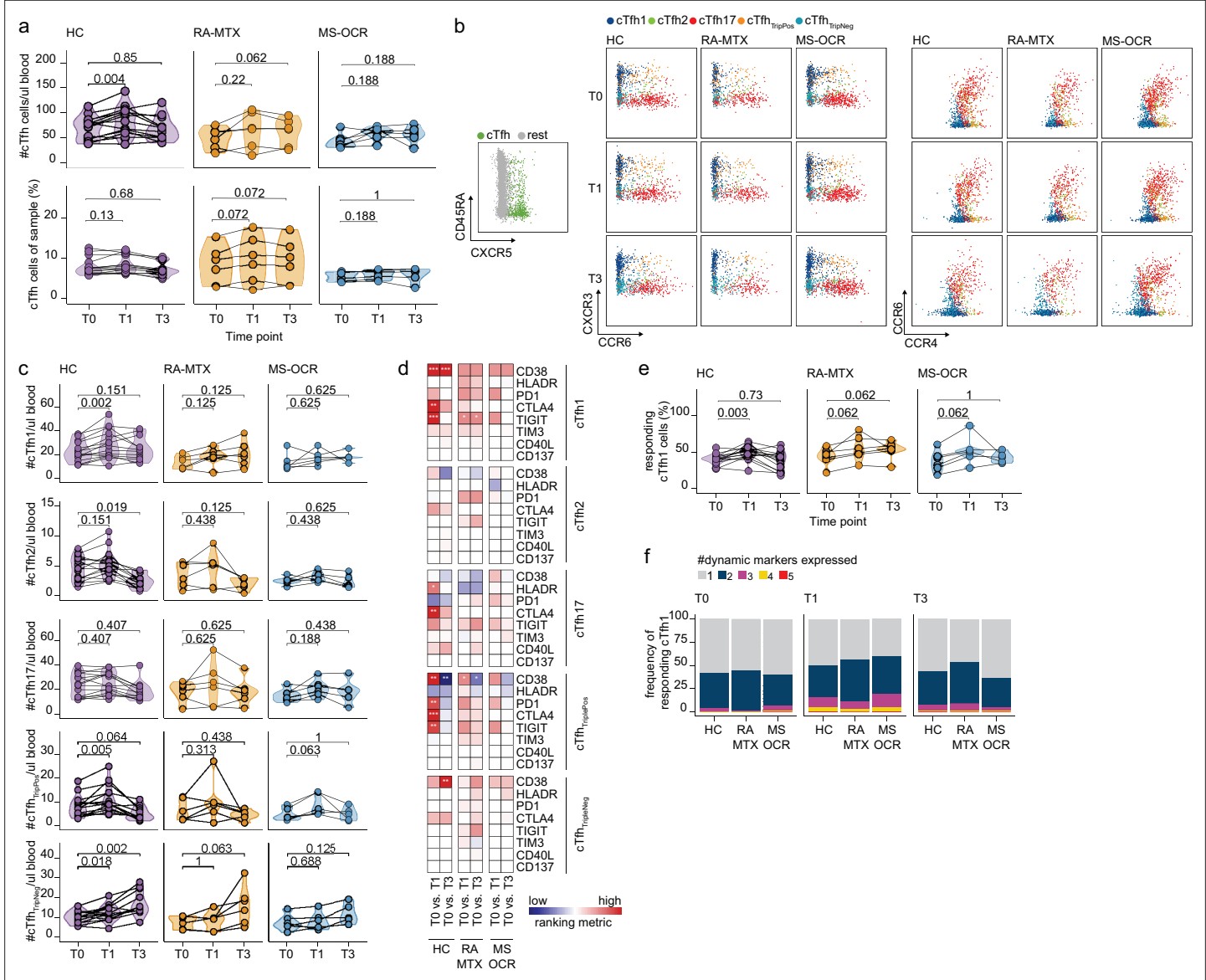

**Figure 3.** Circulating CD4$^+$ T follicular helper (T$_{fh}$) cell responses of severe acute respiratory syndrome coronavirus 2 (SARS-CoV-2) mRNA vaccinated SARS-CoV-2 experienced rheumatoid arthritis using methotrexate (RA-MTX) and multiple sclerosis using ocrelizumab (MS-OCR) patients. (**a**) Count per µl of blood (top) and frequency (bottom) of circulating CD4$^+$ T$_{fh}$ (cT$_{fh}$) cells before (**T0**) and 7 days after first (**T1**) and second vaccination (**T3**). (**b**) Representative flow cytometry plots for the quantification of cT$_{fh}$ cell populations. Colors represent the populations that were identified by unbiased analysis. (**c**) Number of cT$_{fh1}$, cT$_{fh2}$, cT$_{fh17}$, cT$_{fhTriplePos}$, and cT$_{fhTripNeg}$ cells before (**T0**) and 7 days after first (**T1**) and second vaccination (**T3**). (**d**) Heatmap representation showing the overlap in up- and down-regulated dynamic markers expression in cT$_{fh}$ cell subpopulations subsequent first (**T1**) and second (**T3**) vaccination as compared to baseline (**T0**) in the different groups. Each dynamic marker is represented by a single row of colored squares, and each time point-disease group combination is represented by a single column. Color squares represent the dynamic marker expression that is reduced (blue), increased (red), or not affected (white). (**e**) Frequency of responding cT$_{fh1}$ cells before (**T0**) and 7 days after first (**T1**) and second vaccination (**T3**). (**f**) Stacked bar charts representing average fractions of cT$_{fh1}$ cell co-expressing different dynamic molecule combinations. Statistical significance was determined using Wilcoxon signed-rank test (**a, c, d, and e**) with Bonferroni-Holm multiple comparison correction.

The online version of this article includes the following figure supplement(s) for figure 3:

**Figure supplement 1.** Circulating CD4$^+$ T follicular helper (cT$_{fh}$) cells of severe acute respiratory syndrome coronavirus 2 (SARS-CoV-2) mRNA vaccinated SARS-CoV-2 experienced rheumatoid arthritis using methotrexate (RA-MTX) and multiple sclerosis using ocrelizumab (MS-OCR) patients.

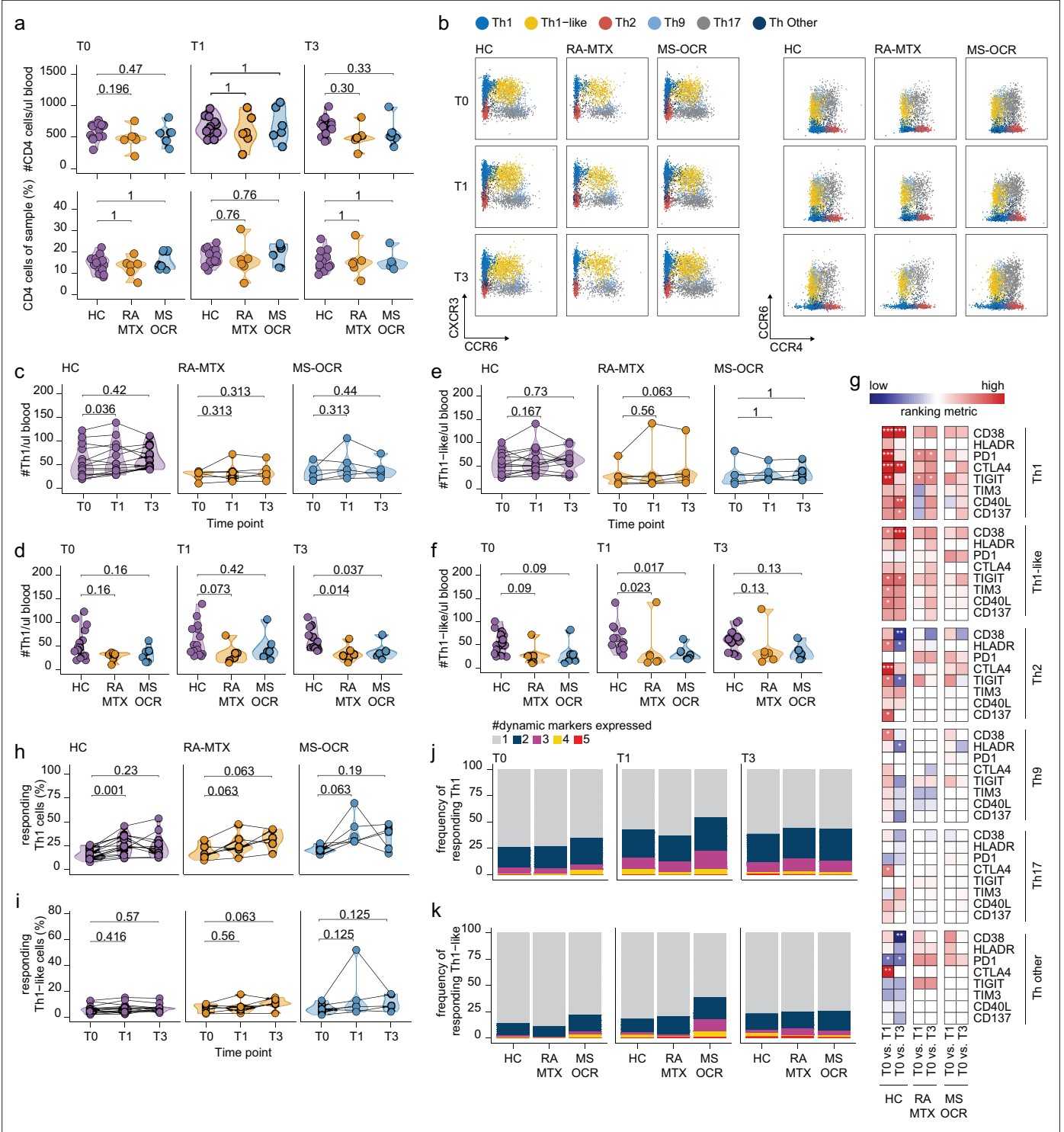

**Figure 4.** Circulating CD4$^+$ T helper (T$_h$) cell responses of severe acute respiratory syndrome coronavirus 2 (SARS-CoV-2) mRNA vaccinated SARS-CoV-2 experienced rheumatoid arthritis using methotrexate (RA-MTX) and multiple sclerosis using ocrelizumab (MS-OCR) patients. (**a**) Count per µl of blood and frequency of CD4$^+$ T helper (T$_h$) cells before (**T0**) and 7 days after first (**T1**) and second vaccination (**T3**). (**b**) Representative flow cytometry plots for the quantification of circulating T$_h$ cell populations. Colors represent the populations that were identified by unbiased analysis. (**c–f**) Count per µl of blood of T$_{h1}$ (**c and d**) and T$_{h1-like}$ (**e and f**) cells before (**T0**) and 7 days after first (**T1**) and second vaccination (**T3**). (**g**) Heatmap representation showing the overlap in up- and down-regulated dynamic markers expression in T$_h$ cell subpopulations subsequent first (**T1**) and second (**T3**) vaccination as compared to baseline (**T0**) in the different groups. Each dynamic marker is represented by a single row of colored squares, and each time point-disease group combination is represented by a single column. Color squares represent the dynamic marker expression that is reduced (blue), increased (red),

*Figure 4 continued on next page*

*Figure 4 continued*

or not affected (white). (**h–i**) Frequency of responding $T_{h1}$ (**h**) and $T_{h\text{-like}}$ cells (**i**) before (**T0**) and 7 days after first (**T1**) and second vaccination (**T3**). (**j–k**) Stacked bar charts representing average fractions of $T_{h1}$ (**j**) and $T_{h1}$-like (**k**) cells co-expressing different dynamic molecules combinations. Statistical significance was determined using a Mann-Whitney test (a, d, and f) and using a Wilcoxon signed-rank test (**c, e, g–i**) with Bonferroni-Holm multiple comparison correction.

The online version of this article includes the following figure supplement(s) for figure 4:

**Figure supplement 1.** Circulating CD4$^+$ T helper ($T_h$) cells of severe acute respiratory syndrome coronavirus 2 (SARS-CoV-2) mRNA vaccinated SARS-CoV-2 experienced rheumatoid arthritis using methotrexate (RA-MTX) and multiple sclerosis using ocrelizumab (MS-OCR) patients.

(*Figure 4h*). In contrast, the responding CD4$^+$ $T_{h1\text{-like}}$ cells remained absent in all groups after both vaccinations (*Figure 4i*). Nevertheless, a greater proportion of CD4$^+$ $T_{h1}$ and $T_{h1\text{-like}}$ responding cells in MS-OCR patients co-expressed three dynamic markers compared to HCs (*Figure 4j–k*, and *Figure 4—figure supplement 1e*). The other helper subsets did not show significant induction of the responding population following either the first or second vaccination (*Figure 4—figure supplement 1d*). Overall, these results indicate that MTX and OCR treatment had a limited effect on the quality of the CD4$^+$ $T_{h1}$ and $T_{h1\text{-like}}$ recall response after the first vaccination.

## MS-OCR but not RA-MTX patients display high-quality CD8$^+$ T cell recall and de novo responses following both vaccinations

CD8$^+$ T cell responses play an important role in viral clearance and reduce disease severity. Therefore, the immune dynamics of various subset of the CD8$^+$ T cell recall in SARS-CoV-2 experienced MS-OCR and RA-MTX patients were elucidated (*Figure 5a* and *Figure 5—figure supplement 1*).

No significant changes in total CD8$^+$ T cells numbers per μl blood were observed in MTX-RA, MS-OCR, and HCs after the first and second vaccination (*Figure 5—figure supplement 1a*), however, a significantly lower total CD8$^+$ T cell number per μl blood was observed in RA-MTX patients compared to HCs (*Figure 5b*). The smaller total CD8$^+$ T cell population in RA-MTX patients was mainly driven by fewer CD8$^+$ $T_{naïve}$ cells compared to HCs throughout the duration of the study (*Figure 5c* and *Figure 5—figure supplement 1c*). In contrast, a trend for a lower proportion and the total number of CD8$^+$ $T_{cm}$ cells per μl blood was observed in MS-OCR patients versus HC (*Figure 5c* and *Figure 5—figure supplement 1c*). Nevertheless, MS-OCR patients display a substantial increase in CD8$^+$ $T_{cm}$ cell count per μl blood and proportions after the first vaccination (*Figure 5—figure supplement 1c*). No significant changes in other phenotypic CD8$^+$ T cell subsets were observed in RA-MTX or MS-OCR patients following either vaccination (*Figure 5—figure supplement 1b, c*).

Next, the effect of OCR and MTX treatment on the response profile of CD8$^+$ T cell subsets was analyzed using the same eight dynamic markers as described for CD4$^+$ $cT_{fh}$ and $T_h$ cells. The expression of these markers was assessed on memory and naive CD8$^+$ T cell populations in SARS-CoV-2 experienced individuals. Differences in the expression of dynamic markers on different CD8$^+$ T cell subsets were observed between RA-MTX, MS-OCR, and HC donors. In RA-MTX patients, a significantly higher proportion of CD8$^+$ $T_{cm}$ cells expressed CD38 after the first vaccination, whereas a more dynamic recall response was observed after the second vaccination where a greater proportion of CD8$^+$ $T_{cm}$ cells exhibited expression of CD38, PD1, TIGIT, and CD40L (*Figure 5d* and *Figure 5—figure supplement 1d*). This was also reflected by the percentage of responding CD8$^+$ $T_{cm}$ cells, which markedly increased after the second vaccination (*Figure 5e*). However, co-expression of three or more dynamic markers was lower in RA-MTX patients versus HCs (*Figure 5f* and *Figure 5—figure supplement 1e*). Together these results indicate a lower quality (exemplified by lack of co-expression of multiple dynamic markers) and/or potentially delayed recall response following vaccination in RA-MTX patients. In contrast, a higher proportion of CD8$^+$ $T_{cm}$ cells in MS-OCR patients displayed expression of CD38, HLA-DR, and TIGIT already after the first vaccination (*Figure 5d*). In addition, a profound increase in the percentage of responding CD8$^+$ $T_{cm}$ cells was observed in MS-OCR patients after the first vaccination, of which substantially more cells co-expressed three to four dynamic markers following first vaccination compared to HCs (*Figure 5f* and *Figure 5—figure supplement 1e*), which is indicative of a high-quality recall response. The CD8$^+$ $T_{cm}$ response observed in MS-OCR patients following the first vaccination largely mimicked that of HCs except that more dynamic markers displayed an increased expression HC (*Figure 5d and e*), which was even more pronounced following the second vaccination in HC (*Figure 5f* and *Figure 5—figure supplement 1e*). This indicates a different dynamic of

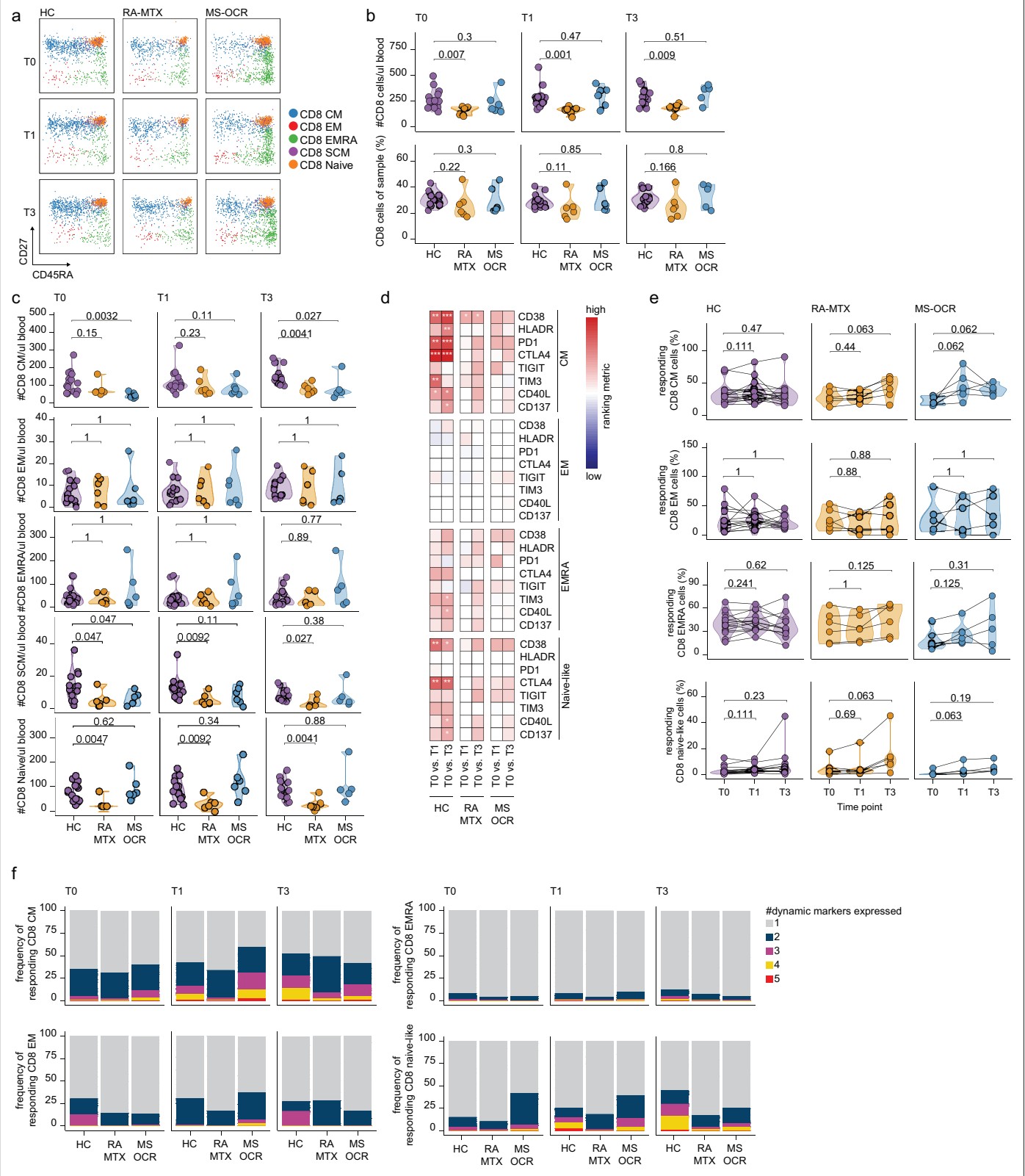

**Figure 5.** CD8+ T cell responses of severe acute respiratory syndrome coronavirus 2 (SARS-CoV-2) mRNA vaccinated SARS-CoV-2 experienced rheumatoid arthritis using methotrexate (RA-MTX) and multiple sclerosis using ocrelizumab (MS-OCR) patients. (**a**) Representative flow cytometry plots for the quantification of circulating CD8+ T cell populations. Colors represent the populations that were identified by unbiased analysis. (**b–c**) Count per μl of blood and frequency of total CD8+ T cells (**a**) and central memory (CM), effector memory (EM), effector memory CD45RA+ (EMRA), stem cell

*Figure 5 continued on next page*

*Figure 5 continued*

memory (SCM), naïve CD8+ T cells (**c**) before (**T0**) and 7 days after first (**T1**) and second vaccination (**T3**) in SARS-CoV-2 experienced RA-MTX patients, MS-OCR patients, and healthy control (HC). (**d**) Heatmap representation showing dynamic marker expression by CD8+ T cell subpopulations that are affected by first (**T1**) and second (**T3**) vaccination as compared to baseline (**T0**) in the different groups. Each dynamic marker is represented by a single row of colored squares, and each time point-disease group combination is represented by a single column. Color squares represent the populations that are significantly reduced (blue) or increased (red) or not affected (white). (**e**) Frequency of CD8 T cell subpopulations before (**T0**) and 7 days after first (**T1**) and second vaccination (**T3**). (**f**) Stacked bar charts representing average fractions of CD8 T cell subpopulations co-expressing different dynamic molecules combinations. Statistical significance was determined using a Mann-Whitney test (**b and c**) and using a Wilcoxon signed-rank test (**d-e**) with Bonferroni-Holm multiple comparison correction.

The online version of this article includes the following figure supplement(s) for figure 5:

**Figure supplement 1.** Circulating CD8+ T cells of severe acute respiratory syndrome coronavirus 2 (SARS-CoV-2) mRNA vaccinated SARS-CoV-2 experienced rheumatoid arthritis using methotrexate (RA-MTX) and multiple sclerosis using ocrelizumab (MS-OCR) patients.

responding memory populations in MS-OCR patients versus HCs. Interestingly, the activation profile of CD8+ $T_{naïve-like}$ cells, which could be indicative of recruitment of de novo $T_{naïve}$ responses or $T_{scm}$ responses, was also different between the patient groups. RA-MTX patients displayed a vast increase in the percentage of responding CD8+ $T_{naïve-like}$ cells after the second vaccination, whereas MS-OCR patients displayed an increase in the percentage of responding CD8+ $T_{naïve-like}$ cells after the first vaccination which remained higher after the second vaccination (*Figure 5e*). However, the CD8+ $T_{naïve-like}$ response in MS-OCR patients was of slightly lower quality compared to HCs, as indicated by the lower co-expression of four and five dynamic markers following both vaccinations (*Figure 5f* and *Figure 5—figure supplement 1e*). Co-expression of more than two dynamic markers was largely absent in the RA-MTX patients at all time points *Figure 5f* and *Figure 5—figure supplement 1a*.

These data demonstrate that especially MS-OCR patients, but not RA-MTX patients, display a strong CD8+ T cell recall response following the first vaccination, while in HCs this recall response is more pronounced following the second vaccination. The effect of both vaccinations on the CD8+ T cell responses in RA-MTX patients seems limited.

## Discussion

The formation of immunological memory following infection and/or vaccination and the subsequent recall response is of great importance to prevent severe disease when re-exposed to the same pathogen. This study aimed to unravel the SARS-CoV-2 immunological recall potential in SARS-CoV-2-experienced patients using two frequently prescribed immunosuppressive medications OCR and MTX. As an alternative to re-exposure with SARS-CoV-2, the immunological recall was investigated following SARS-CoV-2 Moderna mRNA vaccination of SARS-CoV-2 experienced RA-MTX and MS-OCR patients. Results were compared to vaccinated SARS-CoV-2 experienced HCs. Our study provides a comprehensive analysis of the SARS-CoV-2 vaccine-induced recall immune response in SARS-CoV-2 experienced RA-MTX and MS-OCR patients. Despite the small group sizes, both immunosuppressants were shown to affect the vaccine-induced immunological recall response differently.

The quality (measured by the number per µl blood) and quality (measured by the percentage of responding cells and the co-expression of dynamic markers) of the SARS-CoV-2 vaccine-induced CD8+ T cell responses was established. Although MS-OCR patients had a smaller CD8+ $T_{cm}$ population compared to HC before vaccination and upon second vaccination, substantial stronger quality of the CD8+ $T_{cm}$ cell recall response in OCR-treated MS patients was observed compared to those of SARS-CoV-2 experienced HCs following first vaccination. Even though the percentage of responding cells remained high in MS-OCR patients following the second vaccination, HCs displayed responses of slightly higher quality. In general, the overall quality of the CD4+ $T_{fh1}$, $T_{h1}$, and $T_{h1-like}$ recall response was still intact in SARS-CoV-2 experienced OCR-treated MS patients. The dynamic CD4+ $T_{fh1}$ recall response profile was somewhat surprising considering the lack of B cells and significantly smaller CD4+ $T_{fh1}$ population before vaccination. Mice studies previously demonstrated that B cells contribute to the CD4+ T cell activation (*Bouaziz et al., 2007*). However, others also observed CD4+ $T_{fh1}$ cells activation following SARS-CoV-2 vaccination of unexperienced OCR-treated MS patients (*Apostolidis et al., 2021*). One interesting observation was the lack of CTLA-4 expression in SARS-CoV-2 experienced MS-OCR patients. CTLA-4 is known to increase on activated

T cells to inhibit their immune response, which could indicate that by day 7 post vaccination the recall immune response in HCs is already declining (*Chikuma, 2017*). Whether failure to induce CTLA-4 expression on CD4$^+$ T$_{fh1}$ cells in MS-OCR patients is the result of a less tolerant immune system as part of the autoimmune disease or a direct result of missing B cells in the germinal response which may normally be responsible for the induction of CTLA-4 on CD4$^+$ T$_{fh1}$ cells as part of a negative feedback loop remains to be established. In addition, the lack of B cell immunity in MS-OCR patients may have resulted in an altered T$_h$ response, which focused more on the activation of CD4$^+$ T$_{h1}$ responses, skewing the recall immune response to a preserved CD8$^+$ T cell performance. This is in accordance with other studies where low antibody responses following SARS-CoV-2 vaccination of unexperienced MS-OCR patients are also correlated to superior CD8$^+$ T cell responses (*Apostolidis et al., 2021*; *Brill et al., 2021*; *Gadani et al., 2021*; *Madelon et al., 2021*). In contrast to the T cell recall and despite the surprising presence of low SARS-CoV-2 antibody titers before vaccination, anti-CD20 treatment severely impaired the antigen-specific humoral and B cell recall responses following vaccination, which coincidenced with failure to increase antibody titers and/ or seroconversion. OCR treatment is known to eliminate CD20$^+$ B cell populations for prolonged periods (*van Lierop et al., 2022*), which likely hampered the formation of SARS-CoV-2-specific B cell memory in MS patients and indeed memory B cells were undetectable before vaccination. Together, OCR treatment deprived MS patients of an effective B cell recall response, which in part could explain the failure to induce any humoral immunity upon vaccination. It remains to be established why MS-OCR patients seroconverted following the initial infection and if these antibodies have had similar neutralizing qualities as those found in healthy individuals. One possibility is that OCR treatment mainly eradicated circulating CD20$^+$ B cell populations, while B cells in secondary lymphoid organs may have been less affected (*Kamburova et al., 2013*).

MTX treatment in RA patients resulted in delayed immunological recall responses compared to HCs. This was best observed by the antigen-specific humoral immune response following the first vaccination. Despite inducing a rise in antibody titers by day 7, similar levels to SARS-CoV-2 experienced HCs were not reached till day 42 just before the second vaccination. Previous studies observed that de novo antibody titers following SARS-CoV-2 vaccination of MTX-treated RA patients were reduced compared to HCs (*Deepak et al., 2021*; *Furer et al., 2021*; *Haberman et al., 2020*; *Spiera et al., 2021*). Here, we show that recall humoral immunity is merely delayed. MTX is known to hamper the proliferation of lymphocytes (*Cronstein and Aune, 2020*), which could have affected the formation of memory B cells following the initial infection in RA patients and thereby the effectiveness of the recall response, hampering the formation of plasmablasts and plasma cells, which was indeed observed in most RA-MTX patients. Whether delayed humoral recall was the result of lower preexisting B cell memory before vaccination and/or reduced proliferation and differentiation capacity of the B cells, both observed during the course of this study, remains to be established by measuring antigen-specific B cells in these patients over time. However, the immunosuppressive effect of MTX was most striking in the CD8$^+$ T cell compartment. SARS-CoV-2 experienced RA-MTX patients had a significantly smaller total CD8$^+$ T cell population which was mainly driven by a smaller than expected CD8$^+$ T$_{naïve}$ population, whereas their CD8$^+$ T$_{cm}$ population seemed intact. In accordance with the suppressive activity of MTX and in contrast to MS-OCR patients, the memory CD8$^+$ T cell populations in RA-MTX patients did not increase during the course of the study. In addition, responding CD8$^+$ T$_{cm}$ and T$_{naïve}$ populations only increased following the second vaccination. Furthermore, their ability to upregulate the co-expression of multiple dynamic responding markers was severely hampered in CD8$^+$ T$_{cm}$ cells, suggesting that the overall quality of the response was either compromised or delayed, as was observed for antigen-specific humoral recall immunity. A similar effect, albeit less strong, was observed in CD4$^+$ T$_{fh1}$ and T$_{h1}$ cells. Studies that included multiple additional time points are warranted to elucidate the possible explanations.

Of specific interest is the response of CD8$^+$ T$_{naïve}$ cells in MS-OCR patients and HCs following the first vaccination. This indicates that next to immunological recall a novel CD8$^+$ T cell response is generated, which could result in a broader immune response. Further investigation is warranted to establish whether these cells recruit novel clonotypes directed to similar epitopes as during the initial infection or CD8$^+$ T cells directed against additional epitopes. The effect of both vaccinations on the CD8$^+$ T cells response in RA-MTX patients seemed limited, however, it cannot be fully excluded that responses in CD8$^+$ T$_{naïve}$ cells were delayed and not captured by the time points in this study.

Overall, this study demonstrated that MTX and OCR immunosuppressive therapies prevent the induction of a larger-than-expected humoral and cellular recall response that was previously observed in SARS-CoV-2 experienced HCs (*Reynolds et al., 2021*; *Stamatatos et al., 2021*; *Tauzin et al., 2021*; *Wang et al., 2021a*). Despite lacking humoral and B cell immunity, the MS-OCR patients did have a preserved reactive CD8+ T cell recall response. CD8+ T cells are known to form an important second-line defense. Although CD8+ T cells do not prevent infection, they have a key role in viral clearance which contributes to the resolution of symptoms (*Kundu et al., 2022*; *Thevarajan et al., 2020*) and robust CD8+ T cell immunity has been shown to result in overall milder symptoms in SARS-CoV-2 patients (*Rydyznski Moderbacher et al., 2020*; *Tan et al., 2021*). Furthermore, CD8+ T cells recognize conserved epitopes allowing for high cross-reactivity between different SARS-CoV-2 variants (*Alison Tarke et al., 2021*; *Gao et al., 2022*; *GeurtsvanKessel et al., 2021*; *Keeton et al., 2021*; *Liu et al., 2022*; *Rowntree et al., 2022*). While CD8+ T cells immunity will not prevent infection, high proportions of CD8+ T cells have been associated with clinically favorable outcomes (*Kundu et al., 2022*; *Tan et al., 2021*). Ongoing research in this area will be important, particularly in regard to immunosuppressed populations facing novel variants of concern. MTX treatment of RA patients also had impaired recall immune responses after vaccination. Despite a clear delay in the SARS-CoV-2-specific antibody response, further studies including more time points are required to confirm whether the lower response observed in the cellular arm of the immunological recall is delayed or remains impaired over time and how this affects the immune protection of individuals under MTX therapy when infected with new SARS-CoV-2 variants. Together, these findings indicate that SARS-CoV-2 experienced MS-OCR patients may benefit from vaccines that aim to induce a broad CD8+ T cell response. RA-MTX patients lack similar broad-protective cellular immune responses and are likely to benefit more from revaccination strategies with vaccines updated for new variants of concern to induce broad-protective antibody responses.

## Limitations

We recognize that our study had several limitations. First, the analysis performed was limited to the relatively small number of patients available. This particularly prevented correlation. MS and RA patients in general have been careful during the first year of the pandemic not to get infected, so pre-infected patient numbers were relatively scares compared to the general population. Today most patients have been vaccinated before getting infected. Second, although we were able to perform deep-immune profiling on whole blood samples, allowing the detection of vulnerable antibody-secreting B cell populations, this prohibited antigen-specific staining of our immune cells. Instead, we studied a range of dynamic markers which are associated with activated/responding T cells and their co-expression is often associated with higher quality T cell responses (*Koutsakos et al., 2021*; *Thevarajan et al., 2020*). The expression of these markers on antigen-specific cells using AIM peptide stimulations and/or tetramer staining of acute timepoints have been show by others (*Nguyen et al., 2021*; *Oja et al., 2020*; *Rha et al., 2021*, *Habel et al., 2020*, *Geers et al., 2021*, *Mudd et al., 2022*). In addition, we linked the results of our deep-immune profiles to SARS-CoV-2 RBD-specific antibodies measurements. Third, since no MS or RA patients without medication were included, we could not rule out that the disease itself may had an effect on vaccine-induced immune response. Fourth, a comparison with unexperienced RA-MTX and MS-OCR patients and HCs would have been ideal, however, SARS-CoV-2 unexperienced RA-MTX and MS-OCR patients were among the first to be vaccinated under the Dutch national vaccination guidelines and could therefore not be recruited in time for this study. Finally, to understand the delayed dynamics of the cellular responses observed in RA-MTX patients, further studies including multiple time points post vaccination are warranted.

## Acknowledgements

We thank the core facility at Sanquin: Simon Tol and Mark Hoogenboezem for providing technical assistance. We thank Albert Mosselaar, Vesna Melkebeek, and Luca Hensen for technical advice. We would like to thank ZonMw for the funding of the study and the T2B partners, including the patient groups and Health Holland for the support in this study. Funding: This research project was supported by ZonMw (The Netherlands Organization for Health Research and Development, #10430072010007). This project has received funding from the European Union's Horizon 2020 research and innovation program under the Marie Skłodowska-Curie grant agreement (#860003). This study was supported

by the European Commission (SUPPORT-E, #101015756) and by PPOC (#20_21 L2506). CES[b] has received funding from the European Union's Horizon 2020 research, and innovation program under the Marie Skłodowska-Curie grant agreement (#792532). KK is supported by the NHMRC Leadership Investigator Grant to KK (1173871). FE and TWK report (governmental) grants from ZonMw to study immune response after Sars-Cov-2 vaccination in autoimmune diseases.

## Additional information

### Competing interests

Joep Killestein: has speaking relationships with Merck, Biogen, TEVA, Sanofi, Genzyme, Roche and Novartis. AmsterdamUMC, location VUmc, MS Center Amsterdam has received financial support for research activities from Merck, Celgene, Biogen, GlaxoSmithKline, Immunic, Roche, Teva, Sanofi, Genzyme, and Novartis. The other authors declare that no competing interests exist.

### Funding

| Funder | Grant reference number | Author |
| --- | --- | --- |
| ZonMw (The Netherlands Organization for Health Research and Development) | #10430072010007 | Taco W Kuijpers<br>Filip Eftimov<br>S Marieke van Ham |
| European Union's Horizon 2020 research and innovation program under the Marie Skłodowska-Curie grant agreement | #860003 | Taco W Kuijpers<br>Filip Eftimov<br>Theo Rispens |
| European Commission | SUPPORT-E, #101015756 | C Ellen van der Schoot |
| European Union's Horizon 2020 research and innovation program under the Marie Skłodowska-Curie grant agreement | #792532 | Carolien E van de Sandt |
| NHMRC Leadership Investigator Grant | 1173871 | Katherine Kedzierska |
| PPOC | #20_21 L2506 | S Marieke van Ham<br>Anja ten Brinke<br>C Ellen van der Schoot |

The funders had no role in study design, data collection and interpretation, or the decision to submit the work for publication.

### Author contributions

Niels JM Verstegen, Designed the experiments; analyzed the data; provided intellectual input into the study; wrote the manuscript; Ruth R Hagen, Performed the experiments; Jet van den Dijssel, Performed the experiments; Lisan H Kuijper, Performed the experiments; Christine Kreher, Performed the experiments; Thomas Ashhurst, Analyzed the data; provided intellectual input into the study; Laura YL Kummer, Managed the participant database; involved in clinical recruitment; Maurice Steenhuis, Performed the experiments; Mariel Duurland, Performed the experiments; Rivka de Jongh, Performed the experiments; Nina de Jong, Performed the experiments; C Ellen van der Schoot, Provided intellectual input into the study; Amélie V Bos, Provided intellectual input into the study; Erik Mul, Provided intellectual input into the study; Katherine Kedzierska, Provided intellectual input into the study; Koos PJ van Dam, Involved in clinical recruitment; Eileen W Stalman, Involved in clinical recruitment; Laura Boekel, Involved in clinical recruitment; Gertjan Wolbink, Involved in clinical recruitment; Sander W Tas, Involved in clinical recruitment; Joep Killestein, Involved in clinical recruitment; Zoé LE van Kempen, Involved in clinical recruitment; Luuk Wieske, Managed the participant database; involved in clinical recruitment; Taco W Kuijpers, Conceived the study; involved in clinical recruitment; Filip Eftimov, Conceived the study; involved in clinical recruitment; Theo Rispens, Conceived the

study; S Marieke van Ham, Conceived the study; led the study; supervised the study; provided intellectual input into the study; wrote the manuscript; Anja ten Brinke, Supervised the study; provided intellectual input into the study; wrote the manuscript; Carolien E van de Sandt, Supervised the study; designed the experiments; performed the experiments; analyzed the data; provided intellectual input into the study; wrote the manuscript

### Author ORCIDs
Niels JM Verstegen http://orcid.org/0000-0001-5732-4979
Jet van den Dijssel http://orcid.org/0000-0002-4849-0374
Lisan H Kuijper http://orcid.org/0000-0003-3009-0943
Thomas Ashhurst http://orcid.org/0000-0001-7269-7773
Katherine Kedzierska http://orcid.org/0000-0001-6141-335X
Luuk Wieske http://orcid.org/0000-0002-2522-4081
Carolien E van de Sandt http://orcid.org/0000-0002-4155-7433

### Ethics

Human subjects: This study was approved by the medical ethical committee (NL74974.018.20 and EudraCT 2021-001102-30, local METC nummer: 2020_194) and registered at Dutch Trial Register (Trial ID NL8900). Written informed consent was obtained from all study participants when enrolled. Participants were recruited between April 16th 2021 and May 20th 2021 at the MS Center Amsterdam, Amsterdam UMC and the Amsterdam READE Rheumatology and Immunology Center and vaccinated between April 19th 2021 and July 1st 2021 with the mRNA-1273 (Moderna) vaccine at an interval of six weeks, according to the Dutch national vaccination guidelines.

### Decision letter and Author response
Decision letter https://doi.org/10.7554/eLife.77969.sa1
Author response https://doi.org/10.7554/eLife.77969.sa2

---

# Additional files

### Supplementary files
• Supplementary file 1. Characteristics of all participants. Table showing characteristics of participants divided in to immune-mediated inflammatory disorder (IMID) patients with immunosuppressants and healthy controls.

• Supplementary file 2. Whole blood immunophenotyping and CD4 subtyping antibody panels.

• Transparent reporting form

### Data availability
All raw and processed data presented in this study are available at https://flowrepository.org/id/FR-FCM-Z52K.

The following dataset was generated:

| Author(s) | Year | Dataset title | Dataset URL | Database and Identifier |
|---|---|---|---|---|
| Verstegen NJM, Hagen R, van den Dijssel J, Kuijper L, Kreher C, Ashhurst T, Kummer LYL, Steenhuis M, Duurland M, de Jongh R, de Jong N, Bos AV, Kedzierska K, van Dam PJ, Stalman EW, Boekel L, Wolbink GJ, Tas SW, Killestein J, van Kempen ZLE, Wieske L, Kuijpers TW, Eftimov F, Rispens T, van Ham SM, ten Brinke A, van der Sandt CE | 2022 | All raw and processed flow cytometry data | https://flowrepository.org/id/FR-FCM-Z52K | flowrepository, FR-FCM-Z52K |

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
