## [Editor Report]

The article by Verstegen et al. examines humoral and cellular immune response in two subgroups of SARS-CoV-2-experienced immunosuppressed patients receiving two doses of mRNA-1273 vaccine. Further understanding the barriers to seroconversion to COVID-19 vaccination in immunosuppressed populations and how best to overcome these challenges are of great importance. The report is well written, logical, focused and thorough.

---

## [Decision Letter]

**Decision letter after peer review:**

Thank you for submitting your article "Immune dynamics in SARS-CoV-2 experienced immunosuppressed rheumatoid arthritis or multiple sclerosis patients vaccinated with mRNA-1273" for consideration by *eLife*. Your article has been reviewed by 3 peer reviewers, including Sarah Sasson as the Reviewing Editor and Reviewer #1, and the evaluation has been overseen Miles Davenport as the Senior Editor.

Essential revisions:

1) Statistical Analysis: All 3 reviewers have concerns regarding the statistical analysis. Please ensure a Bonferroni correction has been performed where multiple comparisons have been undertaken. This will likely impact on the p-value determination and significance of some results. Please review the validity of a Wilcoxon signed-rank test in Figure 3f.

2) There is little evidence in the current literature that T cell responses alone can prevent SARS-CoV-2 infection. References to this should be modified in the text as highlighted by Reviewer 1.

3) The presented methodology is not sufficient for defining antigen-specific T cells. Contemporary methods for measuring antigen-specific T cells should be performed as highlighted by Reviewer 2 or references to antigen-specific T cells should be removed from the text.

4) The potential confounding effects of the underlying disease processes (that are divergent between groups) should be acknowledged in the Discussion.

*Reviewer #1 (Recommendations for the authors):*

This report is well-written, logically presented and highly focused on two patient populations: MS-OCR and RA-MTX. It provides a thorough mapping of immune responses and recalls adaptive immune responses in SARS-CoV-2 experienced patients undergoing mRNA vaccination. The Figures are clear and concise. Overall, it has the potential to contribute meaningfully to the field.

I have some concerns outlined below that should be addressed prior to publication.

1. Overall, the number of patients in the OCR and MTX is relatively small.

2. In the Abstract>Conclusion the authors state that the expanded CD8^+^ T cell population evident in OCR treated MS patients indicates that this patient group "still benefits" from vaccination. This statement needs to be interpreted in light of the fact that very little data exists to support that T cells alone can protect patients from COVID-19 in real-world clinical settings. This statement should be edited to reflect this complexity.

3. Similarly, the delayed kinetics in MTX lead the authors to state "repeat vaccination strategies should be supported", however, provide no evidence that such repeated strategies are able to overcome treatment with MTX. Indeed, other therapeutic interventions should also be considered and these may include: treatment interruptions, mixed vaccine strategies and synthetic monoclonal antibodies e.g. tixagevimab/cilgavimab.

4. Results: The differences in the CD8^+^ subset in OCR compared to HC appear overstated. In Figure 5e the proportion of CD8^+^ TCM and "naïve-like" cells appear to be incrementally different from HC (p=0.03). It is unclear if Bonferroni corrections have been made for comparisons between groups, but if not, these results are not likely to be statistically significant.

5. In regard to Point 4 the Methods>Statistics do not state a Bonferroni correction has been undertaken for multiple comparisons. This should be done for all analyses in order to determine statistical significance more accurately.

6. Discussion Line 475-479 "Although MS-OCR patients had a smaller CD8^+^ Tcm population compared to HC prior to vaccination, the quantity (measured by the number per µl blood) and quality (measured by the percentage of responding cells and the co-expression of dynamic markers) of the SARS-CoV-2 vaccine-induced CD8^+^ Tcm cell recall response in OCR-treated MS patients outperformed those of SARS-CoV-2 experienced HCs." Again, the comment regarding quantity appears overstated and this sentence should be revised.

7. Line 496-497 "skewing the recall immune response to a superior CD8^+^ T cell performance." Is overstated and not supported by the data. The data is more in keeping with preserved CD8^+^ T cell responses. This sentence should be revised.

8. Line 501 the word "remarkable" should be deleted.

9. Line 554 the word "exceptional" should be removed and replaced with "preserved" or similar.

10. Line 560 "These CD8^+^ T cells are therefore likely to provide MS patients with adequate protection from severe disease following infection with novel SARS-CoV-2 variants." Is highly speculative and not supported by the current literature. This sentence should be removed or revised.

11. Line 567 replace "could" with "may".

12. Line 554 OCR CD8^+^ response described as "exceptional". This should be revised.

13. Line 560-561 Authors postulate high CD8^+^ T cell responses will likely protect against COVID-19 but there is very little data to support this.

*Reviewer #2 (Recommendations for the authors):*

Hagen and colleagues used high-dimensional flow cytometry (38 markers total) to characterise the humoral and cellular immune response to prime-boost vaccination with the mRNA-1273/Moderna SARS-CoV-2 vaccine in healthy controls (HCs), and rheumatoid arthritis (RA) and multiple sclerosis (MS) patients on immunosuppressants, all of which had previously been infected with SARS-CoV-2.

These are important samples to study and this work adds to the body of literature exploring the impact of disease and/or immunosuppressant treatment on the immune response to COVID-19 vaccination. This data will be critical for identifying the best vaccine platforms for these patients, as well as appropriate vaccine regimens.

However, not all the authors' conclusions are supported by the data. For example, there is insufficient evidence to support a "superior recall response of CD8^+^ T central memory cells following first vaccination" in OCR-treated MS patients, as an invalid approach was used to define antigen-specific T cells (cells that upregulated one or more of the following markers relative to the pre-vaccination timepoint: CD38, HLA-DR, PD-1, CTLA-4, TIGIT, TIM-3, CD40L and CD137). Other conclusions need to be rephrased. For example, it is unclear whether "similar CD4^+^ T follicular helper 1 and T helper 1 dynamics" refers to the frequency or activation profile of these cells in OCR-treated MS patients, since Tfh1 cells showed a significant increase in number per μl of blood only in HCs, and not MS or RA patients, following first vaccination.

The main limitation of this paper is the small sample size. Thus, how well the findings will translate to other patients on immunosuppressants is unclear. Disease and treatment are also confounded, so the results could reflect a disease and/or treatment effect. The results indicate the frequency of different cell subsets and the expression of activation markers, but there is a lack of information on the frequency and activation state of antigen-specific T and B cells.

The authors examined the expression of eight dynamic markers (CD38, HLA-DR, PD-1, CTLA-4, TIGIT, TIM-3, CD40L and CD137) "as a surrogate to verify antigen specificity". They have then classed "responding cells" as those expressing one or more of these markers. These criteria are very lenient, and many of the identified cells will not be antigen-specific. To show antigen specificity, the authors could use tetramers or an AIM assay, for example. In Grifoni et al., Cell, 2020, they show that the frequency of SARS-CoV-2-specific CD4^+^ and CD8^+^ T cells in exposed COVID-19 individuals is generally < 1%, much lower than the frequencies of responding cells observed in this paper. The authors should perform one of the suggested assays or remove all references to antigen-specific T cells.

The authors have provided lots of detail in the text about the changes in cell frequencies and marker expression detected or not detected for each cell subset. However, this leads to the key points and the message being somewhat lost. Shortening the text and highlighting the main findings would make the paper more impactful.

The authors compare cell frequencies and dynamic marker expression at T3 (post-vaccine 2) with T0 (pre-vaccine 1), as they did not collect blood pre-vaccine 2 (T2). This makes it challenging to interpret the changes in response to the second vaccine, since cell frequencies and marker expression may differ at T2 relative to T0. The authors should be clear in the text when they are comparing T3 with T0.

In general, the effects of immunosuppressant treatment are confounded by the disease state. The authors should clarify this in the text.

The authors have defined Tfh cells as CXCR5+ CD4^+^ T cells. However, true Tfh cells are located in lymphoid tissues, and are PD-1+ ICOShi and express the transcription factor BCl^-^6 (as discussed in Tangye et al., Nat Rev Immunol, 2013). The authors should rename these populations as Tfh-like or cTfh to distinguish them from true Tfh cells.

What statistical test was used to identify changes in population frequency in Figure 1h,i? Was multiple testing correction performed? Were compositional effects considered in Figure 1i?

The Wilcoxon signed-rank test is not appropriate for Figure 3f.

*Reviewer #3 (Recommendations for the authors):*

The authors have performed a longitudinal study of vaccine responses in 6 individuals with rheumatoid arthritis treated with methotrexate (MTX) and 7 individuals with MS treated with CD20 depletion, all of whom have recovered from COVID-19. Their responses have been compared to 15 reasonably well-matched control individuals.

Their approach is an interesting one – few studies have specifically focused on individuals who have recovered from COVID-19 with these comorbidities. They have chosen to study the Moderna mRNA vaccine, another area where fewer studies have focused. These are all strengths and would add value to the literature as hypothesis-generating for future studies.

The authors have mentioned some of the weaknesses of the study in their conclusions. Namely:

The limited number of participants (further complicated by some of whom hadn't received both vaccine doses).

The fact that they describe their findings as a "recall" response, but this has only been demonstrated using surrogate markers of activation and not in an antigen-specific manner.

To these, I would add the further limitation that:

1. In my opinion, a substantial amount of statistical analysis has been performed on data from a limited number of patients. 13 quite complex multi-panel figures have been created from 13 participants. It is not clear whether any consideration of multiple comparisons has been undertaken.

2. Within the representative flow cytometry plots describing the Tfh subsets there appears to be considerable overlap between the phenotype of some of the subsets (except Tfh1 and Tfh17 which look distinct) which presumably feeds forward into the further analysis of these small populations.

3. I may have missed this in the analysis, but the conclusion regarding CD8 responses in patients with MS – they appear to have a very activated phenotype at T0 (prior to vaccination). Are there actually dynamic differences between baseline at T1 in naive/CM?

4. The study design doesn't allow us to look at the B cell phenotype at timepoint 2 of the RA patients treated with MTX who demonstrate slower kinetics of peak antibody concentration following initial immunisation.

5. One could argue that the study may have been improved if a single disease was studied (e.g. RA where both MTX and anti-CD20 are used rather than introducing a potential confounding effect of the disease process).

Line 116 – there already is a reasonable tranche of literature on this already (both rituximab and methotrexate).

Line 267 – worth mentioning in the methods how many pre-pandemic samples the authors have used to set the threshold for positivity on the serological assay, its sensitivity and specificity.

Using a sufficiently sensitive assay >98% of healthy individuals mount an antibody response to SARS-CoV-2. I found it odd that going into the study a greater percentage of individuals with RA on MTX has a humoral response vs healthy controls. Related to this the authors conclude the patients on MTX have a slower development of antibodies. To justify this, it would also be helpful to specify the precision of the assay, for example the %CV.

Line 271 – It's interesting that some MS patients on anti-CD20 mount humoral responses. What can be said about the relationship between the timing of previous B cell depletion and the timing of that original infection?

Figure 2A and B – I don't agree with the statistical approach to the comparison of timepoints and disease groups here. It looks like a Mann-witney test has been performed that has included the data points below the threshold of the assay. By definition, as the cutoff has been set with pre-pandemic controls, these samples must be negative, as they contain no specific antibodies. They can't be included with a numerical value, even if it is "arbitrary units".

Figure 3A – there does appear to be marked overlap between the defined populations of various Tfh subsets; they don't seem terribly distinct, with the exception of Tfh1 and Thf17. Are there confidence intervals for the gating used to define the various populations here? These populations are then used for further analysis downstream. I'd be concerned about the validity of that analysis.

Figure 5F – I don't fully understand the explanation offered about the differences between the mobilisation of CD8 cells here. For MS patients, they have a usually high frequency of naive-like CD8 T cells expressing activation markers – why is this? Do these significantly increase (i.e mobilised by the vaccine)? Similarly for CD8 CM cells.

Other considerations

– RA and MS are risk factors for severe disease and death from COVID-19; is their survivor bias in the cohort at large?

– There is a wide range of time from SARS-CoV-2 infection to vaccination – is this a potential confounding variable.

– I'd avoid the frequent use of the term "trends" when discussing such a small sample size is likely to be over-interpreting the data.

[Editors' note: further revisions were suggested prior to acceptance, as described below.]

Thank you for resubmitting your work entitled "Immune dynamics in SARS-CoV-2 experienced immunosuppressed rheumatoid arthritis or multiple sclerosis patients vaccinated with mRNA-1273" for further consideration by *eLife*. Your revised article has been evaluated by Miles Davenport (Senior Editor) and a Reviewing Editor.

The manuscript has been substantially improved but there are 3 significant remaining issues that need to be addressed, as outlined below:

1. The role of CD8^+^ T cells in clearing SARS-CoV-2 and improving the clinical course remains overstated in light of the current evidence. Of the supporting literature, the paper by Thevarajan et al. (Nat Med 202) is in fact a case report. Additionally, the paper by Tan et all (Cell Reports 2021) demonstrates an association between CD8^+^ T cell count and milder disease course but evidence of causation cannot be drawn from these.

The Abstract conclusion "Together, these findings indicate that SARS-CoV-2 experienced MS-OCR patients still benefit from vaccination by inducing a broad CD8^+^ T cell response which can contribute to milder disease outcome." Should be edited to: "Together, these findings indicate that SARS-CoV-2 experienced MS-OCR patients may still benefit from vaccination by inducing a broad CD8^+^ T cell response which has been associated with milder disease outcome." (or similar).

Line 770-772 "So even though CD8^+^ T cells will not prevent infection, they are therefore likely to provide MS patients with adequate protection from severe disease following infection with novel SARS-CoV-2 variants." This comment remains speculative and should be edited to: "While CD8^+^ T cells immunity will not prevent infection, high proportions of CD8^+^ T cells have been associated with clinically favorable outcomes. Ongoing research in this area will be important, particularly in regard to immunosuppressed populations facing novel variants of concern." (or similar).

2. Abstract Line 74-76: The statement "A delayed dynamics of vaccine-induced immunological recall in RA-MTX patients support repeated vaccine strategies to protect against future variants of concern, especially for these patients." is not supported by the data. Instead, the authors should highlight the delayed IgG kinetics in the MTX and how this might influence healthcare provider advice to affected patients regarding shielding (mask-wearing etc) and alternative strategies such as treatment interruptions or synthetic IgG products for vulnerable patients. it is unclear how repeated vaccines would impact the kinetics and this has not been shown in this paper.

3. References to "Antigen-specific" T cells have been retained in the manuscript despite no contemporary methods for measuring antigen-specific cells being used. Multi-parameter immunophenotyping is not an adequate replacement for such assays and reporting such data in the context of "antigen specificity" is not scientifically correct in the current context. Such assays can be performed on whole blood (ie the OX40/CD137 AIM assay). Other options include the use of ELISPOT or tetramers both of which could be performed on PBMCs. This data could be retained under a section on advanced immunophenotyping but should not be reported as a measurement of "antigen-specific cells."

---

## [Author Response]

Essential revisions:1) Statistical Analysis: All 3 reviewers have concerns regarding the statistical analysis. Please ensure a Bonferroni correction has been performed where multiple comparisons have been undertaken. This will likely impact on the p-value determination and significance of some results. Please review the validity of a Wilcoxon signed-rank test in Figure 3f.

We have now performed a Bonferroni-Holms correction for all our multiple comparison analysis.

2) There is little evidence in the current literature that T cell responses alone can prevent SARS-CoV-2 infection. References to this should be modified in the text as highlighted by Reviewer 1.

We like to emphasize that we do not claim that CD8^+^ T cells can prevent infection, which is the role of neutralizing antibodies. However, there is sufficient evidence that CD8^+^ T cells contribute to milder disease outcomes, as shown by several studies. We have responded in more detail to the respective questions from Reviewer 1, namely points 2 and 10.

3) The presented methodology is not sufficient for defining antigen-specific T cells. Contemporary methods for measuring antigen-specific T cells should be performed as highlighted by Reviewer 2 or references to antigen-specific T cells should be removed from the text.

We understand the concerns of Reviewer two (point 1) and hope that our answer, including multiple detailed references to previous studies which confirmed the expression of these dynamic markers on antigen-specific T cells, has reassured the Reviewer that dynamic markers were selected with the greatest of care. We have also highlighted some of the challenges of antigen-specific assays and the advantages of our current approach. Unfortunately, no material is available to study the antigen-specific T cell response in an AIMS assay. As we cannot eliminate the slight possibility of bystander activation, we have now rephrased the antigen-specific wording throughout our manuscript by the antigen-induced recall.

4) The potential confounding effects of the underlying disease processes (that are divergent between groups) should be acknowledged in the Discussion.

We have now acknowledged this potential confounding effect in the limitations section of the manuscript.

Reviewer #1 (Recommendations for the authors):This report is well-written, logically presented and highly focused on two patient populations: MS-OCR and RA-MTX. It provides a thorough mapping of immune responses and recalls adaptive immune responses in SARS-CoV-2 experienced patients undergoing mRNA vaccination. The Figures are clear and concise. Overall, it has the potential to contribute meaningfully to the field.

We thank the reviewer for the kind summary and for recognizing the importance of the study. We have addressed the reviewer's comments in a point-by-point form.

I have some concerns outlined below that should be addressed prior to publication.1. Overall, the number of patients in the OCR and MTX is relatively small.

We agree with the reviewer that the number of patients in the OCR and MTX groups is relatively small. Our understanding from clinicians in the field is that both MS-OCR and RA-MTX patients have been very careful prior to vaccination not to get infected, hence the low numbers of infected patients before vaccination. We have attempted to increase our cohort, but patients have been very compliant with the vaccination strategies, hence no additional infected-vaccinated patients could be included in the study. In addition, considering that this group is so rare it will become less likely that any future information on recall responses in this specific pre-infected patient group will be gathered. We review this limitation in the “Limitations” section of the manuscript, which now reads:

“We recognize that our study had several limitations. Firstly, the analysis performed was limited to the relatively small number of patients available. This particularly prevented correlation. MS and RA patients in general have been careful during the first year of the infection not to get infected, so pre-infected patient numbers were relatively low compared to the general population. Today most patients have been vaccinated before getting infected.”

2. In the Abstract>Conclusion the authors state that the expanded CD8^+^ T cell population evident in OCR treated MS patients indicates that this patient group "still benefits" from vaccination. This statement needs to be interpreted in light of the fact that very little data exists to support that T cells alone can protect patients from COVID-19 in real-world clinical settings. This statement should be edited to reflect this complexity.

We apologize for the confusion this sentence may have caused. We agree with the Reviewer that CD8^+^ T cells are unlikely to prevent infection similarly as neutralizing antibodies. Instead, CD8^+^ T cells play a key role in clearing virus-infected cells, which limits viral replication and thereby results contributes to the resolution of the symptoms. In addition, robust CD8^+^ T cells responses have also been shown to result in overall milder symptoms for the patients. This has been demonstrated by several studies including:

Thevarajan *et al.,* 2020, Nat Med; Moderbacher *et al.,* 2020, Cell; Tan *et al.,* 2021, Cell Rep

We have now clarified this in the Abstract, which now reads:

“Together, these findings indicate that SARS-CoV-2 experienced MS-OCR patients still benefit from vaccination by inducing a broad CD8^+^ T cell response which can contribute to milder disease outcome.”

3. Similarly, the delayed kinetics in MTX lead the authors to state "repeat vaccination strategies should be supported", however, provide no evidence that such repeated strategies are able to overcome treatment with MTX. Indeed, other therapeutic interventions should also be considered and these may include: treatment interruptions, mixed vaccine strategies and synthetic monoclonal antibodies e.g. tixagevimab/cilgavimab.

We agree with the Reviewer that multiple therapeutic intervention strategies could be relevant to improving the overall immune response in these patients. However, our study has focussed on their ability to form immune responses upon vaccination. The delayed immune response observed in RA-MTX patients in combination with reduced activation of CD8^+^ T cells, which others have shown to be crossreactive with variants of concern (Geers 2021 Sci Immunol), in combination with their ability to induce antibody responses means that these RA-MTX patients are likely to benefit most from their humoral immunity which needs to be more frequent updated when variants of concern emerge. We have now clarified our statement in the abstract, which now reads:

“A delayed dynamics of vaccine-induced immunological recall in RA-MTX patients support repeated vaccine strategies to protect against future variants of concern, especially for these patients.”

4. Results: The differences in the CD8^+^ subset in OCR compared to HC appear overstated. In Figure 5e the proportion of CD8^+^ TCM and "naïve-like" cells appear to be incrementally different from HC (p=0.03). It is unclear if Bonferroni corrections have been made for comparisons between groups, but if not, these results are not likely to be statistically significant.

We agree with the Reviewer that in the first draft of our manuscript the multiple comparisons corrections were not included. In the current version, we have implemented the Bonferroni-Holms correction for the two comparisons of the disease groups to the healthy control.

5. In regard to Point 4 the Methods>Statistics do not state a Bonferroni correction has been undertaken for multiple comparisons. This should be done for all analyses in order to determine statistical significance more accurately.

We agree with the Reviewer, we have now implemented the Bonferroni-Holms correction throughout the manuscript according to the reviewer’s suggestion.

6. Discussion Line 475-479 "Although MS-OCR patients had a smaller CD8^+^ Tcm population compared to HC prior to vaccination, the quantity (measured by the number per µl blood) and quality (measured by the percentage of responding cells and the co-expression of dynamic markers) of the SARS-CoV-2 vaccine-induced CD8^+^ Tcm cell recall response in OCR-treated MS patients outperformed those of SARS-CoV-2 experienced HCs." Again, the comment regarding quantity appears overstated and this sentence should be revised.

We agree with the Reviewer that the statement with regards to the quantity of the response was stronger than the data would suggest, we have now rephrased this sentence, which now reads:

“The quality (measured by the number per µl blood) and quality (measured by the percentage of responding cells and the co-expression of dynamic markers) of the SARS-CoV-2 vaccine-induced CD8^+^ T cell responses were established. Although MS-OCR patients had a smaller CD8^+^ Tcm population compared to HC before vaccination and upon second vaccination, substantially stronger quality of the CD8^+^ Tcm cell recall response in OCR-treated MS patients was observed compared to those of SARS-CoV2 experienced HCs following the first vaccination.

7. Line 496-497 "skewing the recall immune response to a superior CD8^+^ T cell performance." Is overstated and not supported by the data. The data is more in keeping with preserved CD8^+^ T cell responses. This sentence should be revised.

We agree with the Reviewer and have changed the sentence according to the Reviewer’s suggestion. The sentence now reads:

“In addition, the lack of B cell immunity in MS-OCR patients may have resulted in an altered T_h_ response, which focused more on the activation of CD4^+^ T_h1_ responses, skewing the recall immune response to a preserved CD8^+^ T cell performance.”

8. Line 501 the word "remarkable" should be deleted.

We have deleted the word remarkable from the current manuscript.

9. Line 554 the word "exceptional" should be removed and replaced with "preserved" or similar.

We have replaced the word “exceptional” in the current manuscript with “preserved”. In line with the Reviewer’s suggestions regarding the CD8^+^ T cell response in MS-OCR patients, we have also rephrased the sentence in the abstract, which now reads

“OCR-treated MS patients exhibit a preserved recall response of CD8^+^ T central memory cells following first vaccination compared to healthy controls and a similar CD4^+^ circulating T follicular helper 1 and T helper 1 dynamics, whereas humoral and B cell responses were strongly impaired resulting in absence of SARS-CoV-2 specific humoral immunity.”

10. Line 560 "These CD8^+^ T cells are therefore likely to provide MS patients with adequate protection from severe disease following infection with novel SARS-CoV-2 variants." Is highly speculative and not supported by the current literature. This sentence should be removed or revised.

We respectfully disagree with the Reviewer. We like to emphasize that we do not claim that CD8^+^ T cells can prevent infection, which is the role of neutralizing antibodies. However, multiple studies have shown the importance of CD8^+^ T cell immunity in viral clearance and have demonstrated the key role of strong CD8^+^ T cell immunity before infection and/or during the acute phase of the infection in both resolution of symptoms and milder disease outcome, not only for SARS-CoV-2 (Thevarajan *et al.,* 2020, Nat Med; Moderbacher *et al.,* 2020, Cell; Tan *et al.,* 2021, Cell Rep) but also for other respiratory infections including influenza especially in the absence of neutralizing antibodies (Sridhar 2013 Nat Med; Wang 2015 Nat Comm; Hayward 2015 Am J Resp Crit Care Med). Since MS-OCR patients were unable to produce strong humoral immunity, their protection against severe disease following SARS-CoV-2 infection will greatly depend on their ability to generate other correlates of protection of which CD8^+^ T cells seem to be the most potent. We, therefore, believe that these cells will play an important role in protecting these MS-ORC patients from severe disease in the future, not from the infection itself.

We have now further clarified this by rephrasing the relevant section, which now reads:

“Despite lacking humoral and B cell immunity the MS-OCR patients did have a preserved reactive CD8^+^ T cell recall response. CD8^+^ T cells are known to form an important second-line defense. Although CD8^+^ T cells do not prevent infection, they have a key role in viral clearance which contributes to the resolution of symptoms (Kundu et al., 2022; Thevarajan et al., 2020) and robust CD8^+^ T cell immunity has been shown to result in overall milder symptoms in SARS-CoV-2 patients (Rydyznski Moderbacher et al., 2020; Tan et al., 2021). Furthermore, CD8^+^ T cells recognize conserved epitopes allowing for high crossreactivity between different SARS-CoV-2 variants (Alison Tarke et al., 2021; Gao et al., 2022; GeurtsvanKessel et al., 2021; Keeton et al., 2021; Liu et al., 2022; Rowntree et al., 2021). So even though CD8^+^ T cells will not prevent infection, they are likely to provide MS patients with adequate protection from severe disease following infection with novel SARS-CoV-2 variants.”

11. Line 567 replace "could" with "may".

We have changed the word “could” to “may”.

12. Line 554 OCR CD8^+^ response described as "exceptional". This should be revised.

We agree with the reviewer that the word exceptional may have been too strong in this line. As also touched upon in point 9, we have changed it to “preserved”. The new sentence now reads:

“Despite lacking humoral and B cell immunity the MS-OCR patients did have a preserved reactive CD8^+^ T cell recall response.”

13. Line 560-561 Authors postulate high CD8^+^ T cell responses will likely protect against COVID-19 but there is very little data to support this.

See points 2 and 10.

Reviewer #2 (Recommendations for the authors):Hagen and colleagues used high-dimensional flow cytometry (38 markers total) to characterise the humoral and cellular immune response to prime-boost vaccination with the mRNA-1273/Moderna SARS-CoV-2 vaccine in healthy controls (HCs), and rheumatoid arthritis (RA) and multiple sclerosis (MS) patients on immunosuppressants, all of which had previously been infected with SARS-CoV-2.These are important samples to study and this work adds to the body of literature exploring the impact of disease and/or immunosuppressant treatment on the immune response to COVID-19 vaccination. This data will be critical for identifying the best vaccine platforms for these patients, as well as appropriate vaccine regimens.However, not all the authors' conclusions are supported by the data. For example, there is insufficient evidence to support a "superior recall response of CD8^+^ T central memory cells following first vaccination" in OCR-treated MS patients, as an invalid approach was used to define antigen-specific T cells (cells that upregulated one or more of the following markers relative to the pre-vaccination timepoint: CD38, HLA-DR, PD-1, CTLA-4, TIGIT, TIM-3, CD40L and CD137). Other conclusions need to be rephrased. For example, it is unclear whether "similar CD4^+^ T follicular helper 1 and T helper 1 dynamics" refers to the frequency or activation profile of these cells in OCR-treated MS patients, since Tfh1 cells showed a significant increase in number per μl of blood only in HCs, and not MS or RA patients, following first vaccination.The main limitation of this paper is the small sample size. Thus, how well the findings will translate to other patients on immunosuppressants is unclear. Disease and treatment are also confounded, so the results could reflect a disease and/or treatment effect. The results indicate the frequency of different cell subsets and the expression of activation markers, but there is a lack of information on the frequency and activation state of antigen-specific T and B cells.

We thank the Reviewer for recognizing the importance and relevance of our study. We do agree with the reviewer that due to the small sample size we may have to be a bit more careful with the phrasing that we have used as to not overinterpret or data. We have responded to the Reviewer in a point-by-point form below.

The authors examined the expression of eight dynamic markers (CD38, HLA-DR, PD-1, CTLA-4, TIGIT, TIM-3, CD40L and CD137) "as a surrogate to verify antigen specificity". They have then classed "responding cells" as those expressing one or more of these markers. These criteria are very lenient, and many of the identified cells will not be antigen-specific. To show antigen specificity, the authors could use tetramers or an AIM assay, for example. In Grifoni et al., Cell, 2020, they show that the frequency of SARS-CoV-2-specific CD4^+^ and CD8^+^ T cells in exposed COVID-19 individuals is generally < 1%, much lower than the frequencies of responding cells observed in this paper. The authors should perform one of the suggested assays or remove all references to antigen-specific T cells.

We thank the Reviewer for sharing their concern and hope to reassure the Reviewer that dynamic markers were selected with greatest of care.

As indicated in the limitations section of our manuscript we wished to understand the dynamics of the overall recall response in those patients we performed deep-immune profiling on whole blood samples, allowing the detection of vulnerable antibody-secreting B cell populations, but unfortunately, this prohibited antigen-specific staining of our immune cells, as this is technically challenging for whole blood analysis. Instead, we studied a range of dynamic markers which are associated with antigen induced reactivity and activation and their co-expression is often associated with higher quality T cell responses.

The co-expression of CD38 and HLA-DR is a key phenotype of activation of CD4^+^ and CD8^+^ T cells in response to viral infections, as per reports for Ebola (McElroy 2015 PNAS), influenza including tetramer (Ellebedy 2016 Nat Immunol, Nguyen 2021 Nat Comm) and SARS-CoV-2 (Thevarajan 2020 Nat Med, Koutsakos 2021 Cell Rep Med), co-expression of CD38 and HLA-DR on CD4^+^ and CD8^+^ T cells (assed as the frequency of CD38^+^HLADR^+^ CD8^+^ T cells) rapidly increase during the acute phase of infection and decline over time following resolution of the symptoms and have been previously demonstrated to be directly linked to virus-specific tetramer^+^ CD4^+^ and CD8^+^ T cells (Nguyen 2021 Nat Comm).

The expression of PD-1 has been linked to activation of CD4^+^ T cells during SARS-CoV-2 infection (Thevarajan 2020 Nat Med, Koutsakos 2021 Cell Rep Med), influenza infection (Tetramer+CD4^+^ T cells) (Nguyen 2021 Nat Comm) and vaccination (Koutsakos 2018 Sci Trans Med, Oja 2021 Euro J Immunol) and recently PD-1 was associated with highly functional antigen-specific (Tetramer^+^) CD8^+^ T cells in COVID-19 patients (Rha 2021 Immunity, Oja 2021 Eur J Immunol).

Similar, the combination of CD40L and CD137 has been well established as activation markers of antigen-specific CD4 T cell subsets (Oja 2021 Eur J Immunol) and antigen-specific stimulation of CD8^+^ T cells also increases the expression of CD137 (van de Sandt 2015 J Virol).

These studies have now been cited in the manuscript:

“A T cell activation panel was used to define clusters of highly specialized memory CD4^+^ T cell subsets, CD8^+^ T cell phenotypes, and more in-depth analysis of dynamics markers previously associated with antigen-specific reactivity and activation (Ellebedy et al., 2016; Koutsakos et al., 2021; McElroy et al., 2015; Nguyen et al., 2021; Oja et al., 2020; Rha et al., 2021; Thevarajan et al., 2020), namely CD38, HLA-DR, PD-1, CTLA-4, TIGIT, TIM-3, CD40L and CD137 (Figure 1e and Figure 1—figure supplement 1c,d).”

It is to be expected that other dynamic activation/exhaustion markers can also play a role, however, most studies are limited by their flow cytometry in terms of the number of dynamic markers that can be included in their panels (often limited to 8-14 markers in total). Additional dynamic markers which can be associated with activation or exhaustion, include TIGIT, CTLA-4, and TIM3 which are often studied in relation to various cancer therapies but are frequently overlooked in the virus-specific immune response. This also highlights a unique aspect of our study, as the FACS Symphony allowed us to expand our panels and include these additional markers and study their expression directly ex vivo in combination with the above mentioned and well-defined activation markers without the need for any in vitro manipulation.

We hope that the Reviewer will appreciate that we have carefully selected the dynamic markers based on confirmed antigen-specific markers which were previously confirmed in other studies while exploring the potential of additional markers of interest.

With regards to the AIMS assay, this assay uses either a limited number of peptides or a set of long overlapping peptides which could affect the cleavage of small 7-10 aa peptides and their presentation on HLA-class I molecules needed for CD8^+^ T cell activation, and could therefore result in a slight underestimate of the total antigen-specific CD8^+^ T cell response. Furthermore, T cell responses detected by peptide stimulation and functional readouts could also be affected by differences in antigen presentation. In addition, a recent study (Brodin 2022 Immunity) suggests the potential for bystander activation of T lymphocytes. We have therefore now removed all mention of antigen-specific T cells from the manuscript and replaced those with the antigen-induced recall.

The authors have provided lots of detail in the text about the changes in cell frequencies and marker expression detected or not detected for each cell subset. However, this leads to the key points and the message being somewhat lost. Shortening the text and highlighting the main findings would make the paper more impactful.

We agree with the Reviewer that the level of detail could be overwhelming for the reader. Based on the suggestion from the reviewer (first minor point) we have now simplified our analysis, limited the number of panels in the main figures, and reduced the level of detail in the text by limiting the description of B and T cell population to absolute numbers only which highlight our key findings.

The authors compare cell frequencies and dynamic marker expression at T3 (post-vaccine 2) with T0 (pre-vaccine 1), as they did not collect blood pre-vaccine 2 (T2). This makes it challenging to interpret the changes in response to the second vaccine, since cell frequencies and marker expression may differ at T2 relative to T0. The authors should be clear in the text when they are comparing T3 with T0.

We apologize to the Reviewer that this was not clear in the first draft of the manuscript. In all instances that we make cellular comparisons including T3, we compare to the baseline sample T0. Unfortunately, it was logistically impossible to perform the flow cytometry data acquisition on the day of the second vaccination.

In general, the effects of immunosuppressant treatment are confounded by the disease state. The authors should clarify this in the text.

We agree with the Reviewer that the disease state may by itself also have affected the outcome of the vaccination response, this has now been acknowledged in the limitations section of the manuscript

“Thirdly, since no MS or RA patients without medication were included, we could not rule out that the disease itself may had an effect on vaccine induced immune response.”

The authors have defined Tfh cells as CXCR5+ CD4^+^ T cells. However, true Tfh cells are located in lymphoid tissues, and are PD-1+ ICOShi and express the transcription factor BCl^-^6 (as discussed in Tangye et al., Nat Rev Immunol, 2013). The authors should rename these populations as Tfh-like or cTfh to distinguish them from true Tfh cells.

We agree with the Reviewer that true T follicular helper cells are located in secondary lymphoid tissues. We have renamed all instances of Tfh to cTfh.

What statistical test was used to identify changes in population frequency in Figure 1h,i? Was multiple testing correction performed? Were compositional effects considered in Figure 1i?

In the first draft of our manuscript, no multiple comparison corrections were performed. In the current version of our manuscript, we have implemented the Bonferroni-Holms correction for the comparisons of the two disease groups to the healthy control group.

The Wilcoxon signed-rank test is not appropriate for Figure 3f.

We agree with the Reviewer that Wilcoxon signed-rank test is not appropriate for figure 3f. We have deleted the test from the current manuscript. Due to the small group sizes, we have decided to just show the trends as, this has been addressed as a limitation of our study in the limitations section.

Reviewer #3 (Recommendations for the authors):The authors have performed a longitudinal study of vaccine responses in 6 individuals with rheumatoid arthritis treated with methotrexate (MTX) and 7 individuals with MS treated with CD20 depletion, all of whom have recovered from COVID-19. Their responses have been compared to 15 reasonably well-matched control individuals.Their approach is an interesting one – few studies have specifically focused on individuals who have recovered from COVID-19 with these comorbidities. They have chosen to study the Moderna mRNA vaccine, another area where fewer studies have focused. These are all strengths and would add value to the literature as hypothesis-generating for future studies.The authors have mentioned some of the weaknesses of the study in their conclusions. Namely:The limited number of participants (further complicated by some of whom hadn't received both vaccine doses).The fact that they describe their findings as a "recall" response, but this has only been demonstrated using surrogate markers of activation and not in an antigen-specific manner.To these, I would add the further limitation that:

We thank the Reviewer for recognizing the importance and relevance of our study. We have responded to the Reviewer in a point-by-point form below.

1. In my opinion, a substantial amount of statistical analysis has been performed on data from a limited number of patients. 13 quite complex multi-panel figures have been created from 13 participants. It is not clear whether any consideration of multiple comparisons has been undertaken.

We agree with the Reviewer that in the first draft of our manuscript the multiple comparisons corrections were not included. In the current version, we have implemented the Bonferroni-Holms correction for the comparisons of the two disease groups to the healthy control.

2. Within the representative flow cytometry plots describing the Tfh subsets there appears to be considerable overlap between the phenotype of some of the subsets (except Tfh1 and Tfh17 which look distinct) which presumably feeds forward into the further analysis of these small populations.

Although we agree with the Reviewer that looking at the representative flow cytometry plots the circulating Tfh subsets appear to have considerable overlap. However, for the annotation of the multiple Tfh cells, we have performed an unbiased clustering analysis, of which the results are depicted in figure 1e. This unbiased clustering analysis did place the multiple annotated Tfh cell subset in different meta clusters.

3. I may have missed this in the analysis, but the conclusion regarding CD8 responses in patients with MS – they appear to have a very activated phenotype at T0 (prior to vaccination). Are there actually dynamic differences between baseline at T1 in naive/CM?

The dynamic differences between baseline and 7-10 days after the first vaccination (T1) is summarized in the new figure 5d of the current manuscript. The change in the percentage of CD8^+^ T cells expressing one of the dynamic markers (CD38, HLA-DR, PD1, CTLA-4, TIGIT, TIM3, CD40L or CD137) is represented as a heatmap. All inductions are represented in red, whereas a reduced proportion would be depicted in blue. In the current figure, we significant differences are indicated with a * in this figure. What we hope the Reviewer appreciates is that there is a profound induction of dynamic marker expression in all but the effector memory CD8^+^ T cell subsets.

4. The study design doesn't allow us to look at the B cell phenotype at timepoint 2 of the RA patients treated with MTX who demonstrate slower kinetics of peak antibody concentration following initial immunisation.

Unfortunately, the study design indeed did not allow us to compare the B cell phenotype at timepoint 2 of the RA patients treated with MTX, which would, considering the results presented in the current manuscript, be an interesting time point to follow-up on. This has now been addressed in the limitation section of the manuscript:

“Finally, to understand the delayed dynamics of the cellular responses observed in RA-MTX patients further studies including multiple timepoints post vaccination are warranted.”

5. One could argue that the study may have been improved if a single disease was studied (e.g. RA where both MTX and anti-CD20 are used rather than introducing a potential confounding effect of the disease process).

Studying a single disease may have benefited the statistical analysis as it would have limited the number of multiple comparisons. However, the benefit of studying two disease-medication regiments side by side gave us the ability to highlight their unique differences in immune dynamics following vaccination.

Line 116 – there already is a reasonable tranche of literature on this already (both rituximab and methotrexate).

We have acknowledged the literature that describes this point.

Line 267 – worth mentioning in the methods how many pre-pandemic samples the authors have used to set the threshold for positivity on the serological assay, its sensitivity and specificity.

The serological assay used in this study has been used before (see reference in the text and the URL links below). We have included in the text the number (n=600) of pre-pandemic samples that have been used to set the threshold for positivity.

https://pubmed.ncbi.nlm.nih.gov/34026115/ https://pubmed.ncbi.nlm.nih.gov/33127820/

“Based on IgG titers, the majority of SARS-CoV-2 experienced RA-MTX (67%), MS-OCR patients (50%), and HC (58%) still had detectible antibody titers antibody titers above the cutoff of 4 AU/mL levels; determined using 600 pre-COVID-19 outbreak samples as published before (Steenhuis et al., 2021; Vogelzang et al., 2020) (Figure 2a and Figure 2—figure supplement 1a,b).”

Using a sufficiently sensitive assay >98% of healthy individuals mount an antibody response to SARS-CoV-2. I found it odd that going into the study a greater percentage of individuals with RA on MTX has a humoral response vs healthy controls. Related to this the authors conclude the patients on MTX have a slower development of antibodies. To justify this, it would also be helpful to specify the precision of the assay, for example the %CV.

We find that before vaccination 50% of the individuals had a strong antibody response (Figure 2—figure supplement 1a) and only slightly (not significant) higher in RA-MTX patients. The slower antibody response development observed in RA-MTX patients is mainly based on the continuous increase in Ab titers between T1 and T2 post vaccinations, reading a maximum antibody titer at T2. In contrast, HCs displayed a rapid increase in antibody titer following the first vaccination, reaching maximum titers already at T1 post-vaccination.

We can confirm that the sensitivity of our assay for IgG is >98%. To justify this, we have calculated the precision of the assay. The %CV of this assay is 6,2% using 217 samples.

Line 271 – It's interesting that some MS patients on anti-CD20 mount humoral responses. What can be said about the relationship between the timing of previous B cell depletion and the timing of that original infection?

We agree with the Reviewer that this is a very interesting question we have wondered ourselves. We have checked the timing of the previous B cell depletion and the timing of original infection in the individual patients and did not find a correlation, which may be due to the relatively small size of the patients included in the study. This is also why our group has studied this question in more detail. The paper can be found here (https://pubmed.ncbi.nlm.nih.gov/34847379/), where antibody responses following vaccination of MS patients without prior SARS-CoV-2 infection were studied with regard to the timing of their treatment.

Figure 2A and B – I don't agree with the statistical approach to the comparison of timepoints and disease groups here. It looks like a Mann-witney test has been performed that has included the data points below the threshold of the assay. By definition, as the cutoff has been set with pre-pandemic controls, these samples must be negative, as they contain no specific antibodies. They can't be included with a numerical value, even if it is "arbitrary units".

The threshold set for these assays is based on a statistical analysis of samples considered negative – in this case, pre-outbreak samples. This cut-off itself is therefore also only a statistical reality, representing a set chance (e.g. 95 or 99%) that a 'true negative' sample will yield a signal below the threshold. In other words, the assays do yield numerical values below the threshold, and as such can be included in e.g. comparison of timepoints

Figure 3A – there does appear to be marked overlap between the defined populations of various Tfh subsets; they don't seem terribly distinct, with the exception of Tfh1 and Thf17. Are there confidence intervals for the gating used to define the various populations here? These populations are then used for further analysis downstream. I'd be concerned about the validity of that analysis.

See point 2 above.

Although we agree with the Reviewer that looking at the representative flow cytometry plots the circulating Tfh subsets appear to have considerable overlap. However, for the annotation of the multiple Tfh cells, we have performed an unbiased clustering analysis, of which the results are depicted in figure 1e. This unbiased clustering analysis did place the multiple annotated Tfh cell subset in different meta clusters.

Figure 5F – I don't fully understand the explanation offered about the differences between the mobilisation of CD8 cells here. For MS patients, they have a usually high frequency of naive-like CD8 T cells expressing activation markers – why is this? Do these significantly increase (i.e mobilised by the vaccine)? Similarly for CD8 CM cells.

We are unsure if we understand correctly what the Reviewer means by “mobilization”. Does the Reviewer mean higher expression of dynamic markers on CM and Naïve-like CD8^+^ T cells in MS-OCR patients instead of HCs? We reason that the upregulation of dynamic markers on the CD8^+^ T_CM_ population indicates reactivation of a previously established memory population, most likely during the initial SARS-CoV-2 infection, whereas the upregulation of the T_naive-like_ population most likely indicates the recruitment of de novo CD8^+^ T cells which were not previously activated during natural infection. Indeed, a recent study indicates that vaccination is as a much better ability to activate spike-specific CD8^+^ T cells which are hardly activated during natural infection (Minervina 2022 Nat Immunol). The fact that the frequency of these responses is higher in MS-OCR compared to HCs indicates that they mount a more robust immune response upon vaccination.

Other considerations– RA and MS are risk factors for severe disease and death from COVID-19; is their survivor bias in the cohort at large?

We have cited several studies in the introduction which have studied the risk for severe disease in both RA and MS patients:

“The severity of COVID-19 in individuals on immunosuppressants, varies from mild to severe, depending on the type of immunosuppressants (Gianfrancesco et al., 2020; Möhn et al., 2020; Sparks et al., 2021; Strangfeld et al., 2021) and other underlying risk factors (Baden et al., 2021; F et al., 2021; Hughes et al., 2021; Möhn et al., 2020; Pablos et al., 2020; Sparks et al., 2021; Strangfeld et al., 2021; Williamson et al., 2020; Zabalza et al., 2021).”

– There is a wide range of time from SARS-CoV-2 infection to vaccination – is this a potential confounding variable.

We agree with the Reviewer that there is a wide range between the time of infection and vaccination, however, this wide range is similar between the different groups included in the study. We have included this information to figure 1b as we agree with the Reviewer that this is valuable information to compare the nature and quality of the immune response subsequent vaccination.

– I'd avoid the frequent use of the term "trends" when discussing such a small sample size is likely to be over-interpreting the data.

We thank the reviewer for this observation. In an attempt to focus more on the key messages and less on the trends of less relevant populations we have now excluded the proportion of T and B cell populations from the main figures (data can still be found in the supplemental material) and we focus on the changes in the absolute numbers instead.

[Editors' note: further revisions were suggested prior to acceptance, as described below.]

The manuscript has been substantially improved but there are 3 significant remaining issues that need to be addressed, as outlined below:

We thank the Editor for the additional suggestions, which we have now addressed in the revised version of our manuscript.

1. The role of CD8^+^ T cells in clearing SARS-CoV-2 and improving the clinical course remains overstated in light of the current evidence. Of the supporting literature, the paper by Thevarajan et al. (Nat Med 202) is in fact a case report. Additionally, the paper by Tan et all (Cell Reports 2021) demonstrates an association between CD8^+^ T cell count and milder disease course but evidence of causation cannot be drawn from these.The Abstract conclusion "Together, these findings indicate that SARS-CoV-2 experienced MS-OCR patients still benefit from vaccination by inducing a broad CD8^+^ T cell response which can contribute to milder disease outcome." Should be edited to: "Together, these findings indicate that SARS-CoV-2 experienced MS-OCR patients may still benefit from vaccination by inducing a broad CD8^+^ T cell response which has been associated with milder disease outcome." (or similar).

We agree and the sentence now reads “Together, these findings indicate that SARS-CoV-2 experienced MS-OCR patients may still benefit from vaccination by inducing a broad CD8^+^ T cell response which has been associated with milder disease outcome.” Following Editor’s suggestion.

Line 770-772 "So even though CD8^+^ T cells will not prevent infection, they are therefore likely to provide MS patients with adequate protection from severe disease following infection with novel SARS-CoV-2 variants." This comment remains speculative and should be edited to: "While CD8^+^ T cells immunity will not prevent infection, high proportions of CD8^+^ T cells have been associated with clinically favorable outcomes. Ongoing research in this area will be important, particularly in regard to immunosuppressed populations facing novel variants of concern." (or similar).

We agree and the sentence now reads “While CD8^+^ T cells immunity will not prevent infection, high proportions of CD8^+^ T cells have been associated with clinically favorable outcomes (Kundu et al., 2022; Tan et al., 2021). Ongoing research in this area will be important, particularly in regard to immunosuppressed populations facing novel variants of concern.” Following Editor’s suggestion.

2. Abstract Line 74-76: The statement "A delayed dynamics of vaccine-induced immunological recall in RA-MTX patients support repeated vaccine strategies to protect against future variants of concern, especially for these patients." is not supported by the data. Instead, the authors should highlight the delayed IgG kinetics in the MTX and how this might influence healthcare provider advice to affected patients regarding shielding (mask-wearing etc) and alternative strategies such as treatment interruptions or synthetic IgG products for vulnerable patients. it is unclear how repeated vaccines would impact the kinetics and this has not been shown in this paper.

We appreciate the Editors/Reviewers concerns and we have now adjusted the sentence accordingly, the new sentence now reads “The delayed vaccine-induced IgG kinetics in RA-MTX patients indicate an increased risk after the first vaccination, which might require additional shielding or alternative strategies such as treatment interruptions in vulnerable patients.”

3. References to "Antigen-specific" T cells have been retained in the manuscript despite no contemporary methods for measuring antigen-specific cells being used. Multi-parameter immunophenotyping is not an adequate replacement for such assays and reporting such data in the context of "antigen specificity" is not scientifically correct in the current context. Such assays can be performed on whole blood (ie the OX40/CD137 AIM assay). Other options include the use of ELISPOT or tetramers both of which could be performed on PBMCs. This data could be retained under a section on advanced immunophenotyping but should not be reported as a measurement of "antigen-specific cells."

We understand the Editor and Reviewers concerns regarding antigen-specificity. We have now removed all references to antigen-specific T cells from the manuscript.

Line 353-359 now reads: “A T cell activation panel was used to define clusters of memory CD4^+^ T cell subsets, CD8^+^ T cell phenotypes, and more in-depth analysis of dynamics markers previously associated with activated/responding T cells (Ellebedy et al., 2016; Koutsakos et al., 2021; McElroy et al., 2015; Nguyen et al., 2021; Oja et al., 2020; Rha et al., 2021; Thevarajan et al., 2020), namely CD38, HLA-DR, PD-1, CTLA-4, TIGIT, TIM-3, CD40L and CD137 (Figure 1e and Figure 1—figure supplements 1c-e).”

Line 486-489 now reads: “Next, the functional activation/exhaustion profile of the CD4^+^ cTfh cells upon vaccination was assessed. Expression of CD38, HLA-DR, PD-1, CTLA-4, TIGIT, TIM-3, CD40L, and CD137 (Figure 1—figure supplements 1b) on CD4^+^ cTfh subsets was used to verify activation upon vaccination.”

Line 494-496 now reads: “Analysis of the proportion of responding cells (cells that upregulated one or more dynamic markers) demonstrated that the CD4^+^ cTfh1, cTfh17 and cTfhTripPos subsets are the most responsive after the first vaccination (Figure 3e and Figure 3—figure supplements 1c).”

Line 528-529 now reads: “Next, the response profile of CD4^+^ Th subsets was analyzed using the same eight dynamic markers as described for cTfh cells (Figure 1—figure supplements 1b).”

Line 561-564 now reads: “Next, the effect of OCR and MTX treatment on the response profile of CD8^+^ T cell subsets was analyzed using the same eight dynamic markers as described for CD4^+^ cTfh and Th cells. The expression of these markers was assessed on memory and naïve CD8^+^ T cell populations in SARS-CoV-2 experienced individuals.”

We further clarified the selection of our markers in the limitations section of our manuscript which addresses the limitations of measuring antigen-specific T cell responses in our current analysis:

Line 730-738 now reads:

“Secondly, although we were able to perform deep-immune profiling on whole blood samples, allowing the detection of vulnerable antibody-secreting B cell populations, this prohibited antigen-specific staining of our immune cells. Instead, we studied a range of dynamic markers which are associated with activated/responding T cells and their co-expression is often associated with higher quality T cell responses (Koutsakos et al., 2021; Thevarajan et al., 2020). The expression of these markers on antigen-specific cells using AIM peptide stimulations and/or tetramer staining of acute timepoints have been show by others (Nguyen 2021 Nat Comm, Habel 2020 PNAS, Oja 2020, Rha 2021, Geers 2021 Sci Immunol, Mudd 2022 Cell).”